# VIEW SYNTHESIS WITH SCULPTED NEURAL POINTS

**Yiming Zuo & Jia Deng**
Department of Computer Science, Princeton University
{zuoym,jiadeng}@princeton.edu

## ABSTRACT

We address the task of view synthesis, generating novel views of a scene given a set of images as input. In many recent works such as NeRF (Mildenhall et al., 2020), the scene geometry is parameterized using neural implicit representations (*i.e.*, MLPs). Implicit neural representations have achieved impressive visual quality but have drawbacks in computational efficiency. In this work, we propose a new approach that performs view synthesis using point clouds. It is the first point-based method that achieves better visual quality than NeRF while being $100\times$ faster in rendering speed. Our approach builds on existing works on differentiable point-based rendering but introduces a novel technique we call "Sculpted Neural Points (SNP)", which significantly improves the robustness to errors and holes in the reconstructed point cloud. We further propose to use view-dependent point features based on spherical harmonics to capture non-Lambertian surfaces, and new designs in the point-based rendering pipeline that further boost the performance. Finally, we show that our system supports fine-grained scene editing. Code is available at https://github.com/princeton-vl/SNP.

## 1 INTRODUCTION

We address the task of view synthesis: generating novel views of a scene given a set of images as input. It has important applications including augmented and virtual reality. View synthesis can be posed as the task of recovering from existing images a rendering function that maps an arbitrary viewpoint into an image.

In many recent works, this rendering function is parameterized using neural implicit representations of scene geometry (Mildenhall et al., 2020; Yu et al., 2021c; Park et al., 2021; Garbin et al., 2021; Niemeyer et al., 2021). In particular, NeRF (Mildenhall et al., 2020) represents 3D geometry as a neural network that maps a 3D coordinate to a scalar indicating occupancy. Implicit neural representations have achieved impressive visual quality but are typically computationally inefficient. To render a single pixel, NeRF needs to evaluate the neural network at hundreds of 3D points along the

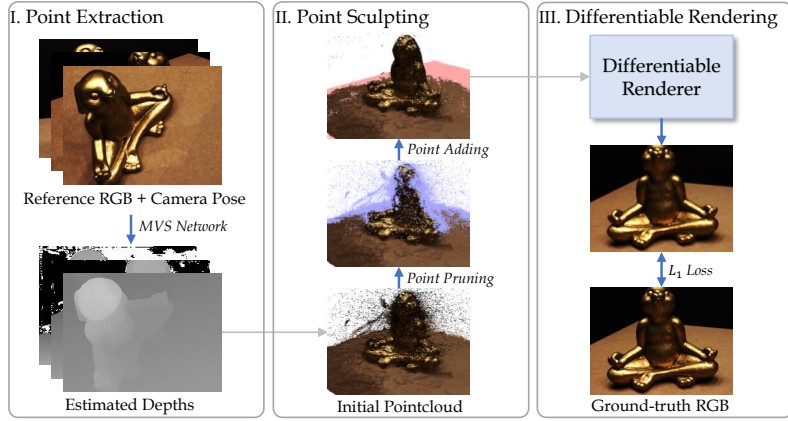

Figure 1: The overall pipeline of the Sculpted Neural Points. We first use an MVS network to extract a point cloud. We then sculpt the point cloud by pruning (blue points) and adding (red points). The featurized point cloud finally passes through a differentiable rendering module to produce the image.

ray, which is wasteful because most of the 3D spaces are unoccupied. NeRF's implicit representation also makes it inflexible for scene editing operations such as deformation, which is important for downstream applications including augmented reality and video games. Several works enable NeRF to do scene editing (Lombardi et al., 2019; Liu et al., 2021; Yang et al., 2021a; Pumarola et al., 2021), but either the way of editing is highly constrained, or images captured under all desired object poses are required.

On the other hand, this limitation is easily overcome by explicit representations such as meshes or point clouds. To rasterize a mesh or a point cloud, no computation is wasted on unoccupied 3D spaces. Scene editing operations such as composition and deformation is also straightforward. Moreover, rasterizing meshes or point clouds is a mature technology already widely deployed in the industry for movies and video games, capable of producing real-time performance and high realism.

An intriguing question is whether we can achieve state-of-the-art visual quality by using explicit representations such as point clouds. The basic framework of point-based neural rendering is to represent the scene as a featurized point cloud, which is reconstructed through a multiview stereo (MVS) system. The features are learned by maximizing photoconsistency on the input images via differentiable rendering. Although this framework has been studied in several recent works (Aliev et al., 2020; Wiles et al., 2020; Lassner & Zollhofer, 2021), the overall rendering quality still lags behind NeRF, mainly due to the ghosting effects and blurriness caused by the errors in geometry.

Our approach adopts this basic framework but introduces a novel technique we call "Sculpted Neural Points (SNP)", which significantly improves the robustness to the errors and holes in the reconstructed point cloud. The idea is to "sculpt" the initial point cloud reconstructed by the MVS system. In particular, we remove existing points and add additional points to improve the photo-consistency of the renders against input images. These sculpting decisions are discrete in nature, but are tightly coupled with gradient-based optimization of the continuous per-point features.

We further propose a few novel designs in the point-based rendering pipeline that boost the performance. We use spherical harmonics (SH) in high-dimensional point feature space to capture non-Lambertian visual effects, which is faster and better than using MLPs. Inspired by Dropout (Srivastava et al., 2014), we propose a point dropout layer that significantly improves the generalization to novel views. Last but not least, we find that it is essential to not use any normalization layers in the U-Net.

Compared to previous works that use point cloud-based representation, ours is the first model that achieves better rendering quality than NeRF, while being $100\times$ faster in rendering, and reducing the training time by 66%. We evaluate our method on common benchmarks including DTU (Jensen et al., 2014), LLFF (Mildenhall et al., 2019), NeRF-Synthetic (Mildenhall et al., 2020), and Tanks&Temples (Knapitsch et al., 2017), and our method shows better or comparable performance against all baselines.

Finally, we show that our model allows fine-grained scene editing in a user-friendly way. Compared to previous works that can only do object-level composition (Lombardi et al., 2019; Yu et al., 2021b; Yang et al., 2021b) or require a special user interface (Liu et al., 2021), our point-based representation inherently supports editing at a finer resolution, and users can use existing graphics toolbox to edit the point cloud without any custom changes.

The main contributions of this paper are three-fold: 1) We propose a novel point-based approach to view synthesis, "Sculpted Neural Points", a technique that is key to achieving high quality and view-consistent output; 2) We demonstrate, for the first time, that a point-based method can achieve better visual quality than NeRF while being $100\times$ faster in rendering. 3) We propose several improvements to the point-based rendering pipeline that significantly boost the visual quality.

## 2 Related Work

Methods for view synthesis can be categorized based on how they represent the scene geometry.

**View Synthesis with Implicit Representations** NeRF (Mildenhall et al., 2020) uses a neural network to map a 3D spatial location to volume density. To render a pixel, the neural network needs to be repeatedly evaluated along the ray, making rendering computationally expensive. Followup works on NeRF (Yu et al., 2021c;b; Park et al., 2021) focus on improving the speed or the general-

ization ability to new scenes or with a limited number of reference views. Our method does not use an implicit representation; instead, we explicitly represent the scene geometry using a point cloud, which allows much faster rendering, as well as easy and flexible scene editing.

**View Synthesis with Point Clouds** To use point clouds for view synthesis, existing approaches typically use an external multiview stereo system to reconstruct a point cloud from the input images, and then optimize the features and 3D positions of each point through differentiable rendering. NPBG (Aliev et al., 2020) is the first work to combine neural rendering with point clouds; it uses a featurized point cloud to represent the scene, and rasterizes with one-pixel point splats at multiple scales followed by a post-processing U-Net to fill the holes. SynSin (Wiles et al., 2020) proposes a soft rasterization pipeline that allows better gradient flow and produces smoother results. Our method achieves significantly better visual quality compared to them. Pulsar (Lassner & Zollhofer, 2021) uses featurized spheres to represent the scene, and proposes a very efficient soft rasterizer that can rasterize millions of spheres in less than 100 milliseconds. Pulsar qualitatively shows that geometry reconstruction can be done through its differentiable rendering system, but has shown no quantitative results on view synthesis. We adopt Pulsar as our backbone.

There are a few concurrent works using point cloud representations. NPBG++ (Rakhimov et al., 2022) focuses on lifting the requirement of per-scene optimization. It does not revise the geometry and is thus more sensitive to the point cloud quality compared to ours. It also proposes to use view-dependent point features, but parameterized as MLP rather than SH as we do. ADOP (Rückert et al., 2022) mainly focuses on unbounded outdoor scenes with large exposure changes among views. Point-NeRF (Xu et al., 2022) uses a featurized point cloud to represent the scene geometry, but renders with volume rendering instead of rasterization.

To revise the initial point cloud provided by an MVS system, existing differential renderers compute gradients with respect to the 3D coordinates of each point. Pulsar (Lassner & Zollhofer, 2021) approximates the gradient by modeling points as spheres with a certain radius, with the color blending weights changing smoothly with respect to the distance of the camera ray to the sphere center. ADOP (Rückert et al., 2022) instead computes the partial derivatives of the photometric loss with respect to the point positions by taking the finite difference in the pixel space. While Point-NeRF (Xu et al., 2022) and our method both refine the point cloud by deleting and adding points, the difference are that: 1) their pruning is based on the volume density optimized for photo-consistency, while our point pruning is based on multiview consistency and doesn't require test-time training; 2) their point growing progressively adds points near existing points, while our point adding has only one round and can add new points in any location.

Our method builds upon existing techniques of differentiable point-based rendering but differs substantially in how we revise the initial point cloud given by an MVS system. Although we find prior methods capable of *local* revision around the existing points by adjusting their locations using the gradients, such local operations, however, cannot fill large holes or add new points in empty spaces far away. In contrast, our sculpting technique is global. It does not use gradients and can add new points in locations arbitrarily far away from existing points.

## 3 APPROACH OVERVIEW

An overview of our approach is illustrated in Fig. 1. The input is a set of $H \times W$ RGB images $\{I_1, \ldots I_m\}$ of $m$ reference views and their corresponding intrinsic and extrinsic camera parameters, $\{C_1, \ldots C_m\}$. We define the camera projection function $\Pi$ and its inverse $\Pi^{-1}$ as follows:

$$\Pi(P, C) := [K_C (R_C P + t_C)]^{\downarrow}; \; \Pi^{-1}(p, d_p, C) := R_c^{-1} \left( K_C^{-1} d_p \begin{bmatrix} p \\ 1 \end{bmatrix} - t_c \right) \tag{1}$$

where $K_C$, $R_C$ and $t_C$ are the intrinsics, rotation, and translation of camera $C$. $P \in \mathbb{R}^3$ is a 3D point and $p \in \mathbb{R}^2$ is its 2D projection in the pixel space with depth $d_p$. Further, $([X, Y, Z]^T)^{\downarrow}$ is defined as $([X/Z, Y/Z]^T)$. Our approach consists of three main components: point cloud reconstruction, point cloud sculpting, and differentiable rendering. In this section, we describe the backbone of our approach with only point cloud reconstruction and differentiable rendering, and leave point cloud sculpting to Sec. 4.

## 3.1 POINT CLOUD RECONSTRUCTION

We use an MVS network (Ma et al., 2022) to extract a dense depth map $\{D_1, \ldots D_m\} \in \mathbb{R}^{\frac{H}{4} \times \frac{W}{4}}$ for each of the reference views. Each depth map is un-projected into a set of 3D points by applying the inverse projection in Eqn. 1. The points from all depth maps are combined, without any filtering, to form a larger set of *original* 3D points $P_o = \{p_1, ..., p_N\}$. We associate point $p_i \in \mathbb{R}^3$ with a learnable $K$-dimensional feature vector $f_i \in \mathbb{R}^K$ and a scalar $o_i \in [0, 1]$ representing its *opacity*.

## 3.2 DIFFERENTIABLE RENDERING

Given a featurized 3D point cloud and a target view, we use a differentiable rendering function with learnable parameters to map the point cloud into an RGB image. For each scene individually, we learn the parameters of this rendering function together with the point features through gradient descent to minimize photometric errors against the input images.

**Spherical Harmonic Point Feature** We use the spherical harmonics (SH) functions to model the view-dependent effects. Recently Yu et al. (2021b;a) brings up the attention of using SH in neural rendering. Unlike previous works that use SH directly in RGB space, we propose to use SH in high-dimensional feature space, where each element of a vector is modulated by a set of SH coefficients.

We use the SH basis up to degree 2, which yields 9 basis in total. This choice follows Yu et al. (2021b) and we find it sufficient to capture highly non-Lambertian surfaces in our experiments. For a 3D point $p_i$, the SH layer takes its feature vector $f_i \in \mathbb{R}^K$ and a target view direction $v$ as input, and outputs a modulated feature vector $s_i \in \mathbb{R}^{\frac{K}{9}}$. Specifically, we first compute the SH basis according to $v$, yielding a basis vector $b_v \in \mathbb{R}^{9 \times 1}$. We then reshape $f_i$ into $f_i' \in \mathbb{R}^{\frac{K}{9} \times 9}$, and finally compute $s_i$ with a dot product $s_i = f_i' \cdot b_v$. Note that evaluating SH functions is cheap as it avoids complex matrix multiplication operations. We find that it leads to better performance and faster rendering speed compared to the MLP parameterization used in NeRF, as shown in Sec. 5.2.

**Differentiable Soft Rasterization** We use soft rasterization proposed in Liu et al. (2019); Lassner & Zollhofer (2021) to convert the view-dependent features into a 2D feature map $F$ given a target view. Soft rasterization blends the features of multiple points hit by a camera ray with weights depending on their depth and opacities. We refer the readers to Pulsar (Lassner & Zollhofer, 2021) for details.

Note that in addition to updating $f_i$, we also compute the gradient of the photometric loss *w.r.t.* the point positions $p_i$ and opacity $o_i$, and optimize them through gradient descent, following Lassner & Zollhofer (2021), which we show helpful in improving fine geometric details in our experiments.

**Point Dropout Layer** We find that the existing point rendering pipeline is prone to over-fitting, *i.e.*, the image quality on test views is much worse than on training views. The reasons are two-fold: **1.** The "training set" for view synthesis consists of only tens of images, and there are barely any data augmentation techniques that can be applied except random cropping. **2.** Some points are covered by their neighbors in training views, but get visible in test views. The features for these points are not well-optimized. The blending weights are very low for these points when the rasterizer is "soft".

To resolve the above issue, we propose to use a "Point Dropout Layer" before rasterization. In each forward pass, we randomly select a subset of points to feed into the rasterizer, whose size depends on the dropout rate $p_d$. Note that at inference time, we cannot simply rasterize all points and multiply the output by $p_d$ as in the neural network (NN) case, because the rasterization operation is non-linear in contrast to the matrix multiplication in NN. Since it is impossible to traverse all subsets, we simply rasterize $L$ multiple random subsets and average the output feature maps at inference time. When rendering videos, we find that sampling independently for each frame causes obvious flickering artifacts. Therefore, we use the same subsets across all frames, which leads to a better consistency.

Intuitively, the point dropout layer allows us to train on an ensemble of point clouds, and give all points a chance to get optimized even if they are covered, and thus alleviating the over-fitting problem. As a side effect, we also gain a speed-up because fewer points get rasterized. Although the design is simple, this idea has not been explored in previous works, and our experiments show that it significantly improves the image quality on test views.

**2-D Rendering without BatchNorm** Given a target view, we convert the 2D feature map $F$ into the RGB image $I_t$ with a 2D ConvNet. We use a U-Net (Ronneberger et al., 2015) with two

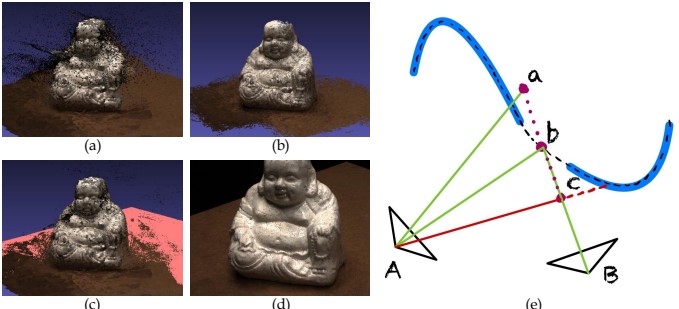

Figure 2: (a) The initial point cloud is incomplete and noisy. (b) The point cloud filtered with Yan et al. (2020) is accurate but incomplete. (c) The output of SNP. It removes most of the outliers and the points added (colored red) further fill the missing areas. (d) The closest training view, which shows what the actual geometry should look like. (e) The **blue** curve and dashed **black** curve represent the reconstructed surface and the actual surface, respectively. A set of candidates is generated along the ray from camera $B$. $c$ is discarded because it occludes the existing surface in view $A$, and $a$ is discarded because we only keep the closest $M = 5$ non-occluding points. Only $b$ is added.

downsampling layers and two upsampling layers, and optionally one more upsampling layer to produce high-resolution outputs. The intuition behind using a U-Net is that it can remove noise in the feature map. We use a dropout layer and relatively small point radius, leaving the rasterized feature map with tiny holes, which makes such denoising necessary. The large receptive field of U-Net is favorable for denoising.

Previous works (Aliev et al., 2020; Rakhimov et al., 2022) directly use the original U-Net design with BatchNorm (Ioffe & Szegedy, 2015) layers, which we find unsuitable for the view synthesis task for two reasons. First, the small training set size makes the estimation of the moving average in BatchNorm unstable. Second, the benefit of accelerated training is minimal since the network is shallow. Therefore, we use no normalization layer in our U-Net.

## 4    POINT SCULPTING

The point clouds reconstructed from MVS usually contain many errors, even with the state-of-the-art MVS systems. The errors typically take the form of distorted or incomplete geometry. If we directly use such point clouds, the synthesized views will have poor visual quality with salient artifacts. To address this issue, we introduce a new technique we call "point sculpting". It has two steps, *Point Pruning* and *Point Adding*. The sculpting procedure and outputs are illustrated in Fig. 2.

### 4.1    POINT PRUNING

The MVS system we use produces a dense depth map for each input image. Like other depth-based systems (Yao et al., 2018; Yan et al., 2020; Chen et al., 2020), it adopts a *fusion* step that merges the depth maps from different views into a final point cloud. A geometry consistency check is often used to remove outlier points. Using the depth maps, the consistency check projects a pixel into another view, reprojects the corresponding point back, and sees if the original pixel is recovered up to a threshold. For example, COLMAP (Schönberger & Frahm, 2016) defines the consistency error $\psi_{\boldsymbol{p}}^{i,j}$ between view $i$ and $j$ for pixel $\boldsymbol{p}$ as:

$$\boldsymbol{q} = \Pi(\Pi^{-1}(\boldsymbol{p}, d_{\boldsymbol{p}}^i, C^i), C^j); \ \psi_{\boldsymbol{p}}^{i,j} = \left\|\boldsymbol{p} - \Pi(\Pi^{-1}(\boldsymbol{q}, d_{\boldsymbol{q}}^j, C^j), C^i)\right\|_2 \tag{2}$$

where $\boldsymbol{q}$ is $\boldsymbol{p}$'s corresponding point in view $j$. Yan et al. (2020) further propose an improved version, Dynamic Consistency Checking (DCC), which achieved the state-of-the-art filtering results. The main problem of this type of forward-backward consistency check is that it tends to be over-aggressive in filtering out points, resulting in highly incomplete geometry that is detrimental to view synthesis. In datasets such as DTU and LLFF, many areas are only visible in a small number of views. Those areas can easily be filtered out by this check as no confident match could be found.

Therefore, we take the raw depth maps from the multiview stereo system and propose a new technique for our own consistency checking geared toward view synthesis. We check only the forward consistency to maximize completeness while still removing outliers. Formally, a pixel $\boldsymbol{p}$ in view $i$ passes the check if and only if $\bigcap_{j=1}^{m} \left[D^j(\Pi^{-1}(\boldsymbol{p}, d_{\boldsymbol{p}}^i, C^i)) \geq \delta_d \cdot d_{\boldsymbol{q}}^j\right]$, where $\boldsymbol{q}$ is $\boldsymbol{p}$'s corresponding

point in view $j$ (same as in Eqn. 2), $d_q^j$ is the predicted depth of $q$ in view $j$, $D^j(\cdot)$ is the depth of a point in view $j$ (the $z$ value of a 3D point in camera $j$'s coordinate), and $\delta_d$ is a hyperparameter controlling the relative tolerance.

Intuitively, our point pruning method keeps a point as long as it is not significantly closer than the original surface to *any* reference view camera. It filters out the points that are floating in the free space between the actual surface and the camera, which are likely to be outliers. It also keeps all points that are only visible in a small number of views. Although the position of such points may not be accurate, it is useful to keep them as candidates for further optimization.

## 4.2 POINT ADDING

As Fig. 2 shows, after pruning, the point cloud can have holes, either due to points being pruned or incorrect depth estimates (e.g. depth estimates that are close to infinity or zero). Previous works (Lassner & Zollhofer, 2021; Yifan et al., 2019) tackled this problem by performing gradient-based updates to the point locations. However, such updates are limited to local changes of existing points and are unable to recover large areas of missing geometry.

We thus introduce a technique to add new points to the pruned point cloud. The basic idea is to find a set of 3D points that, if added to the point cloud, could help minimize the photometric error after optimization of the point features. Note that these new points do not need to be perfect; they just need to be a superset of the ground truth geometry, because the extraneous points can get optimized through the subsequent gradient-based optimization. On the other hand, an excessive number of new points can lead to overfitting and slower rendering, so a good balance is needed. Our point adding algorithm consists of two steps:

- *Optimizing with existing points:* We optimize the features and opacity of the current points through gradient descent until convergence. For the $i$-th input image, we extract an error map between the rendered and ground-truth image: $E_i = ||I_i^{gt} - I_i^{render}||_1$. Note that we use $f_i \in \mathbb{R}^{27}$, which is converted to $s_i \in \mathbb{R}^3$ by the SH layer. $s_i$ is directly treated as RGB values during rasterization, and we use no U-Net in this step, as the U-Net hallucination makes $E_i$ less informative.
- *Proposing new points:* For a pixel $(u, v)$ in an input view $i$, we check if its rendering error $E_i(u, v)$ is bigger than a pre-defined threshold $\delta_e$. If so, we sample 3D points uniformly along the ray of the pixel within the bounds of the scene, and search for points that do not occlude any of the existing points in any of the input views. If multiple such points exist, we propose the closest $M$ points, where $M$ is a hyperparameter. We go through all pixels in all input views, collect all the proposed points, and add them to the existing point cloud.

The design of this algorithm builds upon the assumption that our rendering pipeline can approximate the radiance of each point arbitrarily well on the *input images* and that any high rendering error can only be caused by incomplete geometry, as those areas having no points covered can only take the default background color. Based on this assumption, we propose new points for pixels with high rendering errors, but exclude points that occlude the existing surface in other views. We propose up to $M$ closest points and choose $M = 5$ in our experiments to strike a good balance between geometry coverage and rendering speed. We can alternate between gradient-based optimization and point adding for multiple rounds, but in practice we find one round of point adding to be sufficient for good results. We present the full details of the point adding algorithm in Appendix C.

## 5 EXPERIMENTS

**Datasets** We evaluate our method on DTU (Jensen et al., 2014), LLFF (Mildenhall et al., 2019; 2020), NeRF's Realistic Synthetic 360° (Mildenhall et al., 2020), and Tanks&Temples (Knapitsch et al., 2017). The datasets we choose provide good coverage of both forward-facing and 360° scenes. We evaluate using the standard PSNR, SSIM, and LPIPS (Zhang et al., 2018) metrics.

**Baselines** We compare our model with NeRF (Mildenhall et al., 2020). On DTU and LLFF, we run two point-based methods NPBG (Aliev et al., 2020) and SynSin (Wiles et al., 2020) using the same external MVS system to reconstruct point clouds, with no pruning or adding. We additionally compare against two point-based methods NPBG++ (Rakhimov et al., 2022) and Point-NeRF (Xu et al., 2022), and two voxel-based methods NV Lombardi et al. (2019) and NSVF Liu et al. (2020).

**Implementation Details** We implement our method with PyTorch (Paszke et al., 2019) and Py-Torch3D (Ravi et al., 2020). We experiment on a single RTX 3090 GPU, optimizing for 50,000 steps on each scene with a batch size of 1. We initialize all SH coefficients for each point as 0s and the point opacity as 1. We initialize the U-Net parameters randomly. We set the point radii to be a dataset-specific hyperparameter, which is the same for all points and fixed. See Appendix B for details on the MVS network and other implementations.

## 5.1 PRIMARY RESULTS

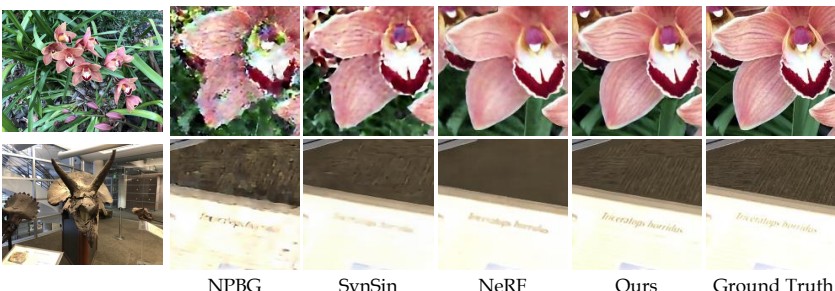

|  | NPBG | SynSin | NeRF | Ours | Ground Truth |

Figure 3: Qualitative comparison of our model v.s. baselines on the LLFF dataset.

**Results on DTU** The quantitative comparison is presented in Tab. 1. Results show that our model has better SSIM and LPIPS, and slightly worse PSNR compared to NeRF. We present the visualizations in Fig. 13, Appendix F. We claim to use LPIPS as the major quality metric, as we find that PSNR and SSIM may not reflect actual visual quality because they are highly sensitive to small pixel shifts (See Appendix. A).

Table 1: Quantitative results on the DTU and the LLFF dataset. For all tables in this paper, we mark the best number in **bold** and the second-best number with an underline.

|  | DTU | | | | LLFF | | | |
| --- | --- | --- | --- | --- | --- | --- | --- | --- |
| Method | NPBG | SynSin | NeRF | **SNP (ours)** | NPBG | SynSin | NeRF | **SNP (ours)** |
| PSNR↑ | 19.38 | 21.04 | **28.97** | 26.68 | 19.98 | 22.34 | **26.50** | 25.32 |
| SSIM↑ | 0.652 | 0.714 | 0.846 | **0.884** | 0.624 | 0.705 | 0.811 | **0.817** |
| LPIPS↓ | 0.412 | 0.337 | 0.266 | **0.156** | 0.454 | 0.351 | 0.250 | **0.229** |

**Results on LLFF** The quantitative results are shown in Tab. 1. Similar to DTU, our method achieves consistently better SSIM and LPIPS, while being slightly worse in PSNR. Qualitative comparisons are shown in Fig. 3 and Fig. 12, Appendix F. Compared to NeRF, our model can reconstruct very fine details. Our method also has significantly better visual quality compared NPBG and SynSin.

**Results on Tanks&Temples** We present the numbers in Tab. 2. All baselines numbers are from Point-NeRF (Xu et al., 2022). Our method achieves comparable quality as Point-NeRF, while being significantly better than other baselines. We present qualitative comparisons in Fig. 16, Appendix F.

Table 2: Quantitative results on Tanks&Temples.

|  | Tanks&Temples | | | | |
| --- | --- | --- | --- | --- | --- |
| Method | NV | NeRF | NSVF | Point-NeRF | **SNP (ours)** |
| PSNR↑ | 23.70 | 25.78 | 28.40 | 29.61 | **29.78** |
| SSIM↑ | 0.848 | 0.864 | 0.900 | **0.954** | 0.942 |
| LPIPS$_{Alex}$↓ | 0.260 | 0.198 | 0.153 | 0.080 | **0.079** |

**Results on NeRF-Synthetic** Results are shown in Tab. 3. Our method achieves comparable performance to NeRF while being worse than Point-NeRF, which is also reflected in Fig. 4. Our method is better at capturing the reflective drum surfaces, while struggles with the microphone which has fine geometry. Our explanation is that our view-dependent point features are very expressive in modeling high-frequency textures, while our point cloud is not accurate enough in cases of fine geometries.

Table 3: Quantitative results on the NeRF-Synthetic dataset. NPBG++ only presents results on the hotdog, ficus, and mic scenes. All other baseline numbers are copied from the Point-NeRF paper.

|  | NeRF-Synthetic (all 8 scenes) | | | | NeRF-Synthetic (3 scenes) | |
| --- | --- | --- | --- | --- | --- | --- |
| Method | NPBG | NeRF | Point-NeRF | **SNP (ours)** | NPBG++ | **SNP (ours)** |
| PSNR↑ | 24.56 | 31.01 | **33.31** | 27.47 | 28.67 | **29.16** |
| SSIM↑ | 0.923 | 0.947 | **0.978** | 0.939 | 0.952 | **0.961** |
| LPIPS↓ | 0.109 | 0.081 | **0.049** | 0.067 | 0.050 | **0.037** |

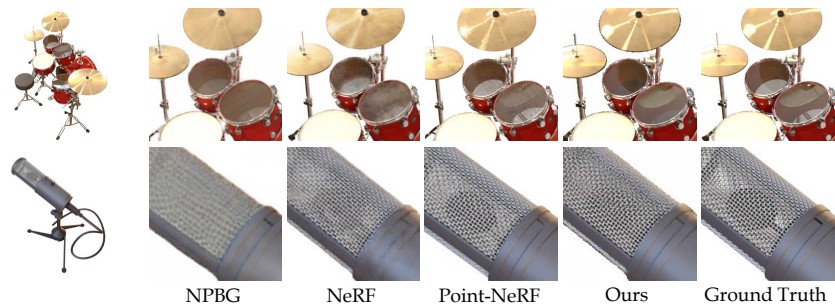

NPBG   NeRF   Point-NeRF   Ours   Ground Truth

Figure 4: Qualitative comparison of our model v.s. baselines on the NeRF-Synthetic dataset.

## 5.2 ABLATION STUDIES

Table 4: Ablation studies on the DTU dataset.

| | View-dependent Layer Latency | Num. Points | Dropout Rate | PSNR↑ | SSIM↑ | LPIPS↓ |
|---|---|---|---|---|---|---|
| Use DCC(Yan et al., 2020) Filtering | 15ms | 3.3M | 50% | 19.97 | 0.844 | 0.196 |
| No Adding; No Pruning | 15ms | 4.2M | 50% | 25.06 | 0.836 | 0.201 |
| No Adding | 15ms | 4.0M | 50% | 26.15 | 0.882 | 0.163 |
| No Gradient-based Refine | 15ms | 4.4M | 50% | 26.52 | 0.880 | 0.157 |
| No View Dependence | N/A | 4.4M | 50% | 25.67 | 0.876 | 0.160 |
| View Dependence w/ MLP | 79ms | 4.4M | 50% | 26.30 | 0.881 | 0.160 |
| No Point Dropout | 31ms | 4.4M | 0% | 25.40 | 0.852 | 0.191 |
| Low Dropout Rate | 23ms | 4.4M | 25% | 26.47 | 0.880 | 0.158 |
| High Dropout Rate | 8ms | 4.4M | 75% | 26.46 | 0.883 | 0.157 |
| BatchNorm in UNet | 15ms | 4.4M | 50% | 25.19 | 0.857 | 0.171 |
| InstanceNorm in UNet | 15ms | 4.4M | 50% | 26.08 | 0.869 | 0.169 |
| 2-layer 1×1 Conv, no UNet | 15ms | 4.4M | 50% | 19.63 | 0.656 | 0.355 |
| Complete Model | 15ms | 4.4M | 50% | 26.68 | 0.884 | 0.156 |

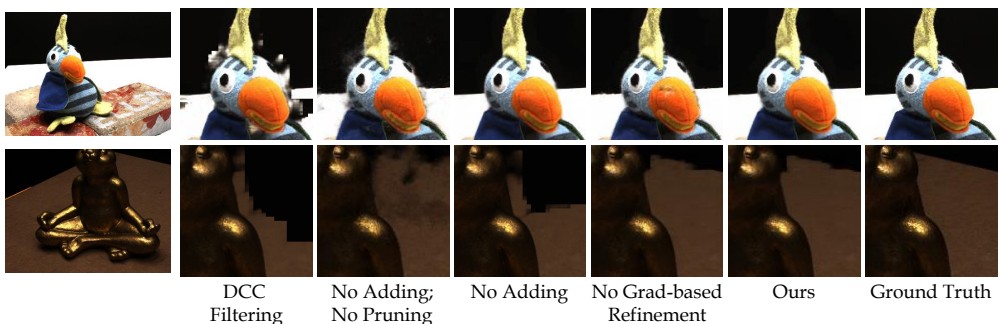

DCC   No Adding;   No Adding   No Grad-based   Ours   Ground Truth
Filtering  No Pruning        Refinement

Figure 5: Qualitative comparison of point sculpting v.s. baselines on the DTU dataset.

We conduct ablation studies of our proposed designs. We show results in Tab. 4. In the 1st block, We compare with several baselines on point cloud refinement, including 1) the filtering algorithm DCC (Yan et al., 2020) which achieves SOTA performance on MVS, 2) using the raw MVS point cloud without any pruning or adding, 3) pruning with the proposed *point pruning* but no *point adding*, 4) using the same point cloud as the complete model, while keeping the point positions and opacity values fixed during gradient updates. Also see Fig. 5 for qualitative comparisons. Results show that all proposed geometry refinement components contribute to the final model. While *point pruning* contributes to sharper object boundaries near the head of the plush, *point adding* is especially helpful for filling large holes on the table in the rabbit scene. See Fig. 11, Appendix C for visualizations of the point cloud generated by each method.

The 2nd block shows that using SH reduces the layer latency by 82% and improves the PSNR by 0.38dB compared to MLP. For the MLP baseline, we use a 2-layer MLP with 256 hidden units, which takes as input the concatenation of the point feature and the positional-encoded view direction, following NeRF. See also Fig. 14, Appendix F for visualization of the non-Lambertian effect learned by our model. The 3rd block shows that using dropout layer improves the PSNR by 1.28dB, and the model is not sensitive to the dropout rate. The 4th block shows that not using any normalization

layers improves the PSNR by $1.49$dB compared to using BatchNorm, and by $0.60$dB compared to InstanceNorm. Replacing U-Net with a 2-layer $1{\times}1$ Conv network gives significantly worse results.

## 5.3 SCENE EDITING

We show that our system supports, with high fidelity, scene editing operations such as scene composition, object deformation, and erasing. Results are shown in the inset figure on the right. Composition is achieved by first co-training two scenes with separate point features and a shared U-Net, then

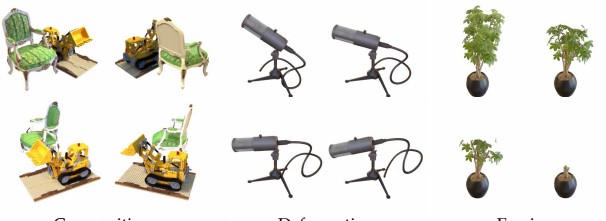

Composition       Deformation       Erasing

putting the points into a single scene at inference time. For deformation, we export the sculpted point cloud into MeshLab (Cignoni et al., 2008), where we manually select the moving part and its axis of rotation. For erasing, we filter out points based on their $z$ coordinates.

Compared to existing neural rendering pipelines that support scene editing, our system has two main advantages: **1. Fine-grained editing**: Previous works (Lombardi et al., 2019; Yu et al., 2021b; Yang et al., 2021b) use explicit representations like voxel grids, which are typically limited in resolution. Therefore, they can only achieve object-level operations such as composition. In comparison, we represent object surfaces densely with millions of points, so we can do fine-grained editing such as object deformation. **2. Ease of use**: Previous works doing scene editing with NeRF either require a special interface to take user inputs (Liu et al., 2021), or a complex pipeline that uses meshes as an intermediate representation (Yuan et al., 2022). In contrast, our point cloud representation is directly supported by nearly all graphics toolboxes such as MeshLab or Blender, which allows users to edit the scene intuitively without any specialized tool.

## 5.4 INFERENCE SPEED, TRAINING TIME, AND MODEL SIZE

We compare our model's inference speed, training time, and model size with a few baselines on the NeRF-Synthetic dataset, shown in Tab. 5. All speeds are benchmarked using an RTX 3090 GPU. Compared to NeRF, our model is more than $100\times$ faster in inference and requires only 33% training time. PlenOctrees (Yu et al., 2021b) bakes the radiance field into a voxel-based cache, resulting in faster rendering speed but also a significantly larger model size and longer training time. NPBG (Aliev et al., 2020) achieves faster inference speed with their one-pixel point splats, but at the cost of worse visual quality. Finally, we are about $25\times$ faster than Point-NeRF (Xu et al., 2022) in rendering while other metrics are roughly the same.

Table 5: On the NeRF-Synthetic dataset, we compare model inference speed, training time, model size, and rendering quality (measured in LPIPS) with baselines.

|  | NeRF | PlenOctrees | NPBG | Point-NeRF | **SNP (ours)** |
|---|---|---|---|---|---|
| Inference↑ (FPS) | 0.053 | 127 | 20.3 | 0.192 | 5.06 |
| Training↓ (Hours) | 20 | 50 | 6.9 | 8.0 | 6.6 |
| Model Size↓ | 14MB | 1.9GB | 31MB | 106MB | 290MB |
| LPIPS↓ | 0.081 | 0.053 | 0.109 | 0.049 | 0.067 |

## 6 DISCUSSIONS AND LIMITATIONS

There are a few limitations that need to be addressed in future work: **1) MVS dependency**. Although the proposed point sculpting can partly solve this problem, the performance of the system still depends heavily on the MVS quality. That said, as MVS systems continue to improve, we do not see this as a fundamental limitation in the long run. **2) View Consistency**. Our system has a 2D U-Net and is thus only approximately 3D consistent. Especially when viewed in videos, some background areas have flickering effects due to the hallucination of U-Net. Doing away with a 2D post-processing network is a future direction. **3) Far-away background**. Our current system cannot deal with outdoor scenes with arbitrarily far-away objects (*e.g.* sky or clouds). Using a spherical environment map as in Zhang et al. (2020); Rückert et al. (2022) could resolve this problem.

## ACKNOWLEDGMENTS

This work was partially supported by the National Science Foundation under Award IIS-1942981. We thank Jing Wen, Zeyu Ma, and Lahav Lipson for their insightful discussions. We thank Artem Sevastopolsky for generously sharing the NPBG data and clarifying the paper details.

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

APPENDIX

## A  LIMITATION OF PSNR AND SSIM AS VIEW SYNTHESIS METRICS

PSNR, SSIM, and LPIPS are commonly used metrics to measure the similarity between two images. The LPIPS paper (Zhang et al., 2018) found that LPIPS is significantly more robust than PSNR or SSIM under distortions such as random noise, blurring, and spatial shifts. Although the evidence is pretty strong for the 2D cases, the robustness of the three metrics is seldom discussed in the context of view synthesis, where 3D geometry plays a key role.

We here provide a detailed case study to show that LPIPS is more robust and consistent with human perception than PSNR and SSIM for the novel view synthesis task, and is more suitable as an evaluation metric.

### A.1  MISALIGNMENT IN NOVEL VIEW SYNTHESIS

Our key observation is that spatial misalignment is a very common type of error in the view synthesis task. The misalignment could be caused by inaccurate camera intrinsics and extrinsics (e.g., the COLMAP camera pose used in LLFF). Even in the DTU dataset, where the cameras are carefully calibrated with a robotic arm, there is misalignment caused by ambiguity in geometry. For example, many background objects are only visible in one view, which makes it impossible to synthesize pixels that are identical to the ground truth.

Such errors in the camera poses of existing views will cause small shifts in the rendering of a new target view. We find that PSNR and SSIM, especially PSNR, can drop significantly in the presence of such pixel shifts, even though they have no detectable impact on the perceived visual quality. In contrast, LPIPS is relatively more robust. In Fig. 6, we show by a synthetic experiment that a slight shift could cause a huge drop in PSNR, while LPIPS is robust to such a shift, behaving more similarly to the human perceptual system, which is very *insensitive* to a such shift. See the figure caption for details.



(a) 7.17/0.095/**0.242**  (b) 7.54/**0.237**/0.401  (c) Ground Truth

Figure 6: The numbers under each image are PNSR, SSIM, and LPIPS. Based on the reference image (c), we create (a) by shifting 1 pixel, and (b) by adjusting the pixel intensities toward grey. Clearly (a) is visually more similar to the reference image, while only LPIPS agrees with the fact.

We also conduct an experiment on the NeRF-Synthetic dataset to demonstrate our findings, where the ground-truth camera poses are perfect. We apply a tiny noise $\sim \mathcal{N}(0, 0.01\mathbf{I})$ in the tangent space to the camera rotation. As shown in Tab. 6, under such a small perturbation, the PSNR and SSIM of our method and NeRF become even lower than the NPBG baseline, which has significantly worse visual quality, while our LPIPS score is consistently better. In Fig. 7, we show the visual effect of such perturbation.

Table 6: We apply an almost imperceptible random noise $\sim \mathcal{N}(0, 0.01\mathbf{I})$ to the camera rotation in the tangent space. While PSNR and SSIM are sensitive to such perturbation, LPIPS remains stable.

|  | NeRF | Ours | NPBG | NeRF | Ours |
|---|---|---|---|---|---|
| Camera Noise | ✓ | ✓ | ✗ | ✗ | ✗ |
| PSNR↑ | 24.21 | 22.70 | 24.56 | 31.01 | 27.47 |
| SSIM↑ | 0.887 | 0.876 | 0.923 | 0.947 | 0.939 |
| LPIPS↓ | 0.089 | 0.088 | 0.109 | 0.081 | 0.067 |

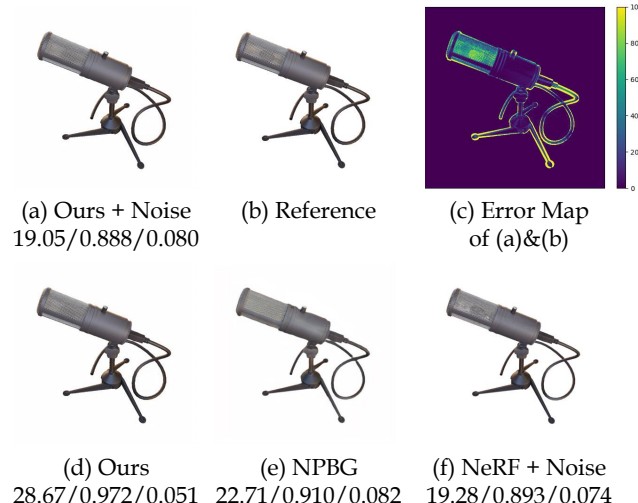

(a) Ours + Noise          (b) Reference          (c) Error Map
19.05/0.888/0.080                                of (a)&(b)

(d) Ours              (e) NPBG            (f) NeRF + Noise
28.67/0.972/0.051   22.71/0.910/0.082   19.28/0.893/0.074

Figure 7: The numbers under each image are PSNR, SSIM, and LPIPS. We see from (a) and (b) that the tiny perturbation of camera poses is perceptually undetectable, unless visualized as the error map (c). Under perturbation, our method achieves better visual quality compared to NPBG (e), which is only reflected by LPIPS, but not PSNR or SSIM.

## A.2 QUALITATIVE EVALUATION

We show qualitative results on the LLFF dataset in Fig. 8. Our rendering results have better visual quality and contain sharper details compared to NeRF, while having lower PSNR due to the misalignment caused by imperfect camera parameters. We further apply a $3 \times 3$ Gaussian blurring to our results, which leads to worse visual quality but higher PSNR. It indicates that blurred images could have advantages in PSNR under misalignment, whereas LPIPS always prefers our sharper results.

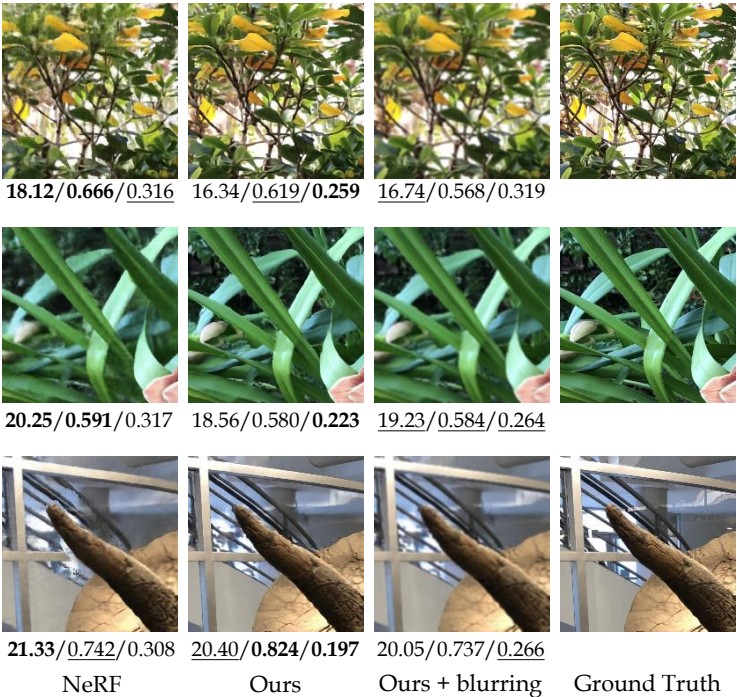

**18.12**/**0.666**/0.316    16.34/0.619/**0.259**    16.74/0.568/0.319

**20.25**/**0.591**/0.317    18.56/0.580/**0.223**    19.23/0.584/0.264

**21.33**/0.742/0.308    20.40/**0.824**/**0.197**    20.05/0.737/0.266
NeRF                  Ours             Ours + blurring       Ground Truth

Figure 8: The numbers under each image are PNSR, SSIM, and LPIPS towards the reference image. See text for details.

## B IMPLEMENTATION DETAILS

### B.1 DATASETS

**DTU** We select 10 scenes from the DTU test set. We only use scenes from the test set because the training data of CER-MVS (Ma et al., 2022) includes the DTU training scenes. The render resolution is $400 \times 300$, following PixelNeRF (Yu et al., 2021c). We reserve 1 in every 7 images for testing, resulting in 42 training views and 7 test views.

**LLFF** LLFF contains 8 forward-facing scenes. Following NeRF, we render under the resolution of $1008 \times 720$ and reserve $1/8$ of the views for testing.

**NeRF-Synthetic** We use the same setting as NeRF (Mildenhall et al., 2020). The training set and test set for each scene contain 100 and 200 images respectively, and the resolution is $800 \times 800$.

**Tanks&Temples** We test on 5 scenes using the split and masks provided by NSVF (Liu et al., 2020). The render resolution is $1088 \times 640$, following Point-NeRF (Xu et al., 2022).

### B.2 MVS RECONSTRUCTION

We use the CER-MVS (Ma et al., 2022) network to extract depth maps for all scenes. We scale the scenes so that the median depth is about 600, which is the depth scale that CER-MVS was trained on. For the DTU dataset, we use the CER-MVS trained on the DTU training set, which has no overlap with our test scenes. For the NeRF-Synthetic and the Tanks&Temples datasets, we use the CER-MVS trained on BlendedMVS (Yao et al., 2020).

For LLFF, since the domain gap is large, we take the CER-MVS pre-trained on DTU and finetune the model individually for each scene with the dense depth map provided by COLMAP (Schönberger & Frahm, 2016; Schönberger et al., 2016). Specifically, we use COLMAP customized by the authors of NerfingMVS (Wei et al., 2021), which additionally generates a confidence mask based on geometric consistency. We apply loss only to the areas with a positive mask. The intuition for finetuning is that we leverage COLMAP at geometric consistency areas, and we rely on the learning-based priors at the areas where COLMAP fails, thus taking advantage of both methods. To justify the proposed point cloud generation pipeline, we compare our method to directly using the COLMAP point cloud, as shown in Tab. 7 and Fig. 9. The "w/ COLMAP" baseline uses the fused point cloud from COLMAP, with our differentiable rendering but no point sculpting. Results show that our method achieves better performance, especially in texture-less areas such as the ceiling, where the COLMAP point cloud is incomplete.

Running the COLMAP MVS takes 0.3 hours. We finetune for 5k steps on each scene, which takes about 1 hour. The 1.3 hours additional overhead is insignificant compared to the 6-8 hours training time of our model. Taking this overhead into account, our method is still more efficient in training than NeRF (about 21 hours on LLFF). All numbers are reported on a single RTX 3090 GPU.

After the point sculpting step, we downsample the point cloud sizes to 500K for LLFF and NeRF-Synthetic, and 1M for DTU and Tanks&Temples, to make the training and inference speed roughly the same for all scenes.

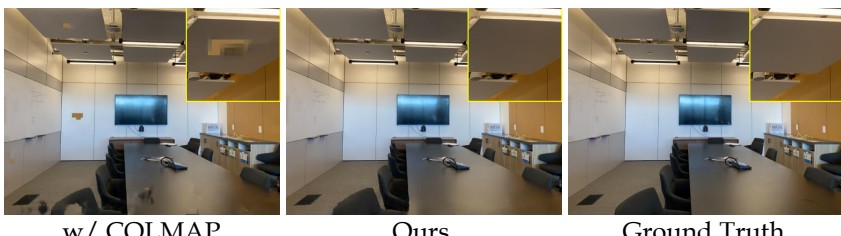

| w/ COLMAP | Ours | Ground Truth |

Figure 9: We visually compare the results of using the COLMAP fused point cloud against our proposed MVS finetuning + point sculpting. Our point cloud has higher completeness and thus better visual quality, especially in texture-less areas (*e.g.,* the zoomed-in ceiling region).

Table 7: Quantitative comparison with the COLMAP point cloud baseline.

|  | PSNR↑ | SSIM↑ | LPIPS↓ |
|---|---|---|---|
| w/ COLMAP point cloud | 24.81 | **0.817** | 0.240 |
| Ours | **25.32** | **0.817** | **0.229** |

## B.3 MODEL DETAILS

The feature vector $f_i$ attached to each point is 288-dim, and the modulated feature outputted by the SH layer is 32-dim. Unlike using the tiny 8-dim feature vector as in NPBG (Aliev et al., 2020), we find that increasing the feature dimension would monotonically give better results. We choose the dimension by balancing the performance and speed/memory cost.

For the point dropout layer, we use a dropout rate $p_d = 0.5$. At inference time we rasterize $L = 2$ random subsets for all experiments. We empirically find that $L = 2$ gives a significant improvement in performance over $L = 1$, while the advantage is minimal for averaging more subsets. See Appendix E for details.

The U-Net we use has two down-sampling layers and two up-sampling layers with skip connections between the feature maps with the same resolution. Empirically, we find that limiting the capacity of the U-Net could reduce artifacts and leads to better generalization to novel views. Therefore, we use a shallower network compared to the 5-layer U-Net used in NPBG (Aliev et al., 2020). For the LLFF, NeRF-Synthetic, and Tanks&Temples datasets which have higher resolution, we rasterize at half resolution and add an additional up-sampling layer to output the high-resolution images. The U-Net is randomly initialized and trained individually for each scene.

We use an $L_1$ loss between the rendered image and the target. We also add a total variation loss on the 2D feature map to improve the smoothness of the output:

$$L_{TV}(F) = \sum_{i,j} |F_{i+1,j} - F_{i,j}| + |F_{i,j+1} - F_{i,j}| \qquad (3)$$

The final loss can be written as $L = L_1 + \lambda_{TV} \cdot L_{TV}$, where $\lambda_{TV}$ is set to 0.01 in all experiments. The learning rates for the U-Net, the feature vectors $f$, the point position $p$, and the opacity $o$ are set to $10^{-4}$, $10^{-2}$, $10^{-4}$, $10^{-4}$, respectively. The rasterization "softness" hyper-parameter $\gamma$ is selected to be $10^{-3}$. The opacity is passed through a sigmoid layer to map its range into $[0, 1]$.

We use the Adam optimizer (Kingma & Ba, 2014) with the OneCycleLR learning rate scheduler (Smith & Topin, 2019), and train the model with a batch size of 1 for 50,000 steps for all experiments. No augmentation including random cropping is applied. We find using a larger batch size and data augmentations are not helpful.

We set the point radius to be $1.5 \times 10^{-3}$, $1.0 \times 10^{-3}$, $7.5 \times 10^{-3}$ for all scenes in DTU, LLFF and NeRF-Synthetic, respectively. For Tanks&Temples, since there is a large scale variance among scenes, we use different radius for each scene. Specifically, radius are $1.0 \times 10^{-3}$ for "Ignatius" and "Family", $1.0 \times 10^{-2}$ for "Truck" and "Caterpillar", and $5.0 \times 10^{-3}$ for "Barn".

## C THE POINT SCULPTING ALGORITHM

For point pruning, we set the depth tolerance threshold to $\delta_d = 0.8$ in all experiments.

The formal description of the point adding algorithm is presented in Alg. 1. The definition of $\Pi$, $\Pi^{-1}$, $R_C$ and $t_C$ are the same as Eqn. 1 in the main body of the paper.

For DTU, we set $z_{near}$ and $z_{far}$ to be 800 and 1400, respectively. For LLFF, since those are unbounded scenes, we borrow the idea of using inverse depth as the NDC parameterization in NeRF. We sample uniformly in the inverse depth space, and the equivalent $z_{near}$ and $z_{far}$ is 800 and $+\infty$. For NeRF-Synthetic, we use $z_{near} = 2.0$ and $z_{far} = 6.0$, and for Tanks&Temples we find the depth range for each scene using the bounding box provided by the NSVF (Liu et al., 2020) authors.

$z_{step}$ is set so that there are 100 depth bins. The hyperparameters $\delta_e$ is set to $5 \cdot Avg(E)$, as we find that using $\delta_e$ relative to the average error improves the robustness. We keep the closest $M = 5$

points for each ray. Fig. 11 shows the visualization of the point cloud generated by DCC (Yan et al., 2020), the initial MVS point cloud ("No Adding; No Pruning"), the point cloud filtered by our pruning only ("No Adding"), and our point sculpting method. We see that most of the outlier points floating in free spaces are pruned by our method (blue points in "Pruning Only"). We also show how many points get pruned and added in Tab. 8. Note that the initial numbers of points for each scene are slightly different because we filter out the points based on the scene depth range $z_{near}$ and $z_{far}$. More points are added for *scan110*, *scan114*, and *scan118* (corresponding to the last 3 rows in Fig. 11), where the tables in the scene contain large holes in the initial point cloud.

Table 8: Number of points statistics for pruning and adding.

Number of Points ($\times 10^5$)

| | | | | | Scan Id. | | | | | | |
|---|---|---|---|---|---|---|---|---|---|---|---|
| | 1 | 4 | 15 | 24 | 32 | 33 | 49 | 110 | 114 | 118 | Avg. |
| Initial | 42.93 | 39.53 | 43.51 | 45.20 | 42.61 | 45.30 | 43.64 | 37.42 | 39.42 | 40.11 | 41.97 |
| Pruning | -1.75 | -2.22 | -0.86 | -1.10 | -2.71 | -3.89 | -1.87 | -1.60 | -1.38 | -1.94 | -1.93 |
| Adding | 1.34 | 0.80 | 1.31 | 1.32 | 1.71 | 1.60 | 1.42 | 8.87 | 8.93 | 11.50 | 3.88 |

## D  COMPARISON WITH PULSAR

Pulsar (Lassner & Zollhofer, 2021) is a highly related work to our method. Unfortunately, we are unable to do a quantitative comparison with Pulsar, because the code for the view synthesis part of Pulsar is not available. We show qualitatively that our method is better than Pulsar in Fig. 10, where the Pulsar figure is directly copied from the paper. Conceptually, while using the same rasterization backbone, our method is better because 1) we use MVS to initialize point positions, whereas they start from random positions; 2) we do point adding to improve the completeness, whereas they don't; 3) we use SH features and a point dropout layer to boost the performance, whereas they don't have such designs.

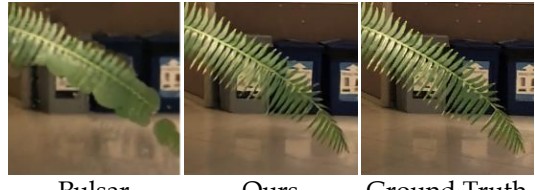

Pulsar            Ours            Ground Truth

Figure 10: Qualitative comparison with Pulsar on the *fern* scene of the LLFF dataset. Our method is much better at capturing the fine details such as the fern leaves.

## E  ANALYSIS OF THE POINT DROPOUT LAYER

We do an analysis of the visual quality v.s. the number of subsets used in the point dropout layer. We do experiments on the DTU dataset and the results are shown in Tab. 9. Results show that averaging two subsets gives a significant improvement over using only one subset, but using more subsets doesn't help. On the other hand, the latency grows almost linearly as we rasterize more subsets. Therefore, one would prefer using a small number of subsets such as 2.

Table 9: Performance v.s. the number of subsets used. Results are averaged on 10 DTU scenes.

| | Number of Subsets | | | | |
|---|---|---|---|---|---|
| | 1 | 2 | 3 | 4 | 5 |
| PSNR↑ | 26.49 | 26.68 | 26.66 | 26.70 | 26.70 |
| SSIM↑ | 0.879 | 0.884 | 0.885 | 0.886 | 0.885 |
| LPIPS↓ | 0.159 | 0.156 | 0.156 | 0.157 | 0.157 |
| FPS↑ | 6.2 | 3.4 | 2.3 | 1.8 | 1.4 |

---

**Algorithm 1** Point Adding

---

1: **Input** Differentiable renderer $R$, Initial point cloud $P_o$, Cameras $\{C_{1:m}\}$,
2: Reference images $\{I_{1:m}^{ref}\}$, Depth maps $\{D_{1:m}\}$, Scene near bound $z_{near}$,
3: Scene far bound $z_{far}$, Depth sampling stride $z_{step}$, Error map threshold $\delta_e$,
4: Max candidates to keep for each ray $M$.
5: **Output** Updated point cloud $P_o$
6:
7: Train model $R$ with current $P_o$ on reference views until convergence.
8: Render images $\{I_{1:m}^{pred}\}$ and compute error maps $\{E_{1:m} = ||I_{1:m}^{pred} - I_{1:m}^{ref}||_1\}$.
9: Candidates = $\emptyset$
10: **for** $i = 0 : m$ **do**            ▷ loop over the views
11:     **for** $u = 0 : H, v = 0 : W$ **do**            ▷ loop over all pixels
12:         **if** $E_i(u, v) \geq \delta_e$ **then**            ▷ propose in the regions with large error
13:              Candidates = SAMPLECANDIDATES($C_i$, $(u, v)$)
14:              counter = 0
15:             **for** c **in** Candidates **do**
16:                 **if** EVALUATECANDIDATES($c$, $C_{1:m}$, $D_{1:m}$) == 1 **then**
17:                      $P_o \leftarrow P_o \cup c$            ▷ add to the point cloud
18:                      counter = counter + 1
19:                 **end if**
20:                 **if** counter $\geq$ M **then**
21:                     **break**            ▷ break once we have $M$ candidates
22:                 **end if**
23:             **end for**
24:         **end if**
25:     **end for**
26: **end for**
27:
28: **function** SAMPLECANDIDATES($C$, $(u, v)$)
29:     Candidates = $\emptyset$
30:     **for** $z = z_{near} : z_{step} : z_{far}$ **do**            ▷ linearly sample the depth
31:          Candidates $\leftarrow$ Candidates $\cup \Pi^{-1}([u, v]^T, z, C)$            ▷ add the 3D point
32:     **end for**
33:     **return** Candidates
34: **end function**
35:
36: **function** EVALUATECANDIDATES($c$, $C_{1:m}$, $D_{1:m}$)
37:     no_conflict = 1
38:     **for** $i = 0 : m$ **do**            ▷ loop over the views
39:          $(u_i, v_i) = \Pi(c, C_i)$            ▷ the corresponding 2D point
40:          $[x_i, y_i, z_i]^T = R_{C_i} c + t_{C_i}$            ▷ $c$ in the $i$-th camera coordinates
41:         **if** $0 \leq u_i \leq H - 1, 0 \leq v_i \leq W - 1$ **and** $z_i < D_i(u_i, v_i)$ **then**
42:              no_conflict = 0            ▷ conflict if $c$ occludes the existing surface
43:         **end if**
44:     **end for**
45:     **return** no_conflict
46: **end function**

---

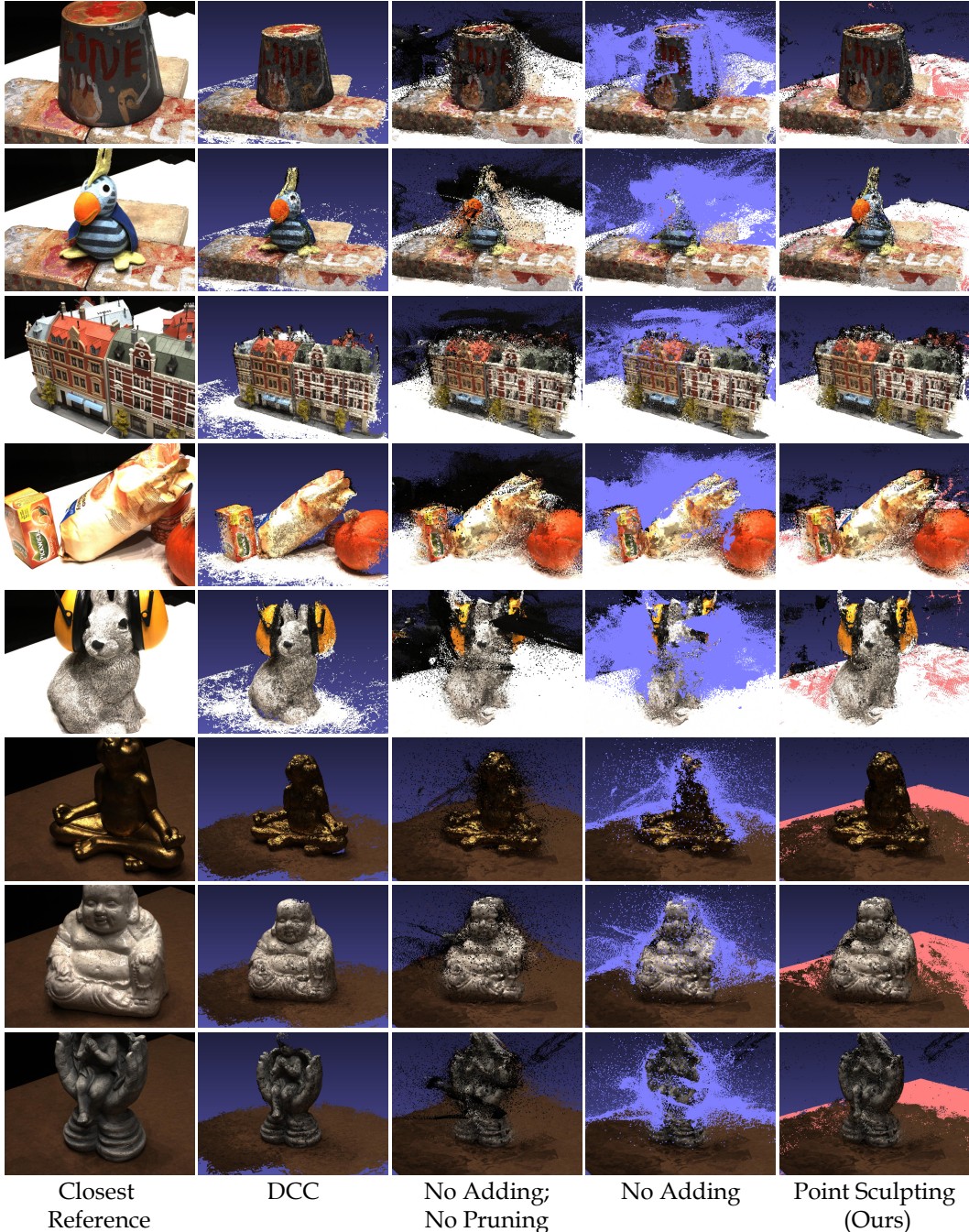

| Closest Reference | DCC | No Adding; No Pruning | No Adding | Point Sculpting (Ours) |

Figure 11: Qualitative comparison of different point cloud refinement algorithms. Compared to the initial point cloud ("No Adding; No Pruning"), our point sculpting method removes the outliers (the blue ones in the "No Adding" column), and adds new points (the red ones) that further fill the missing areas. Although DCC (Yan et al., 2020) provides point clouds with higher accuracy in the foreground, the completeness is much worse than our results. For scenes where the initial point cloud quality is low (*e.g.,* the bottom 3 rows), more points are used to fill the holes. **Although the final sculpted point cloud can still contain outliers and small holes, those can be handled by the gradient-based refinement.**

## F  MORE VISUALIZATIONS

Due to the limited space in the main body of the paper, we present more visualizations of our model in this section, to help the readers better compare our methods with the baselines. Results are presented in Fig. 12, 13, 14, 15, and 16. See figure captions for detailed comparisons.

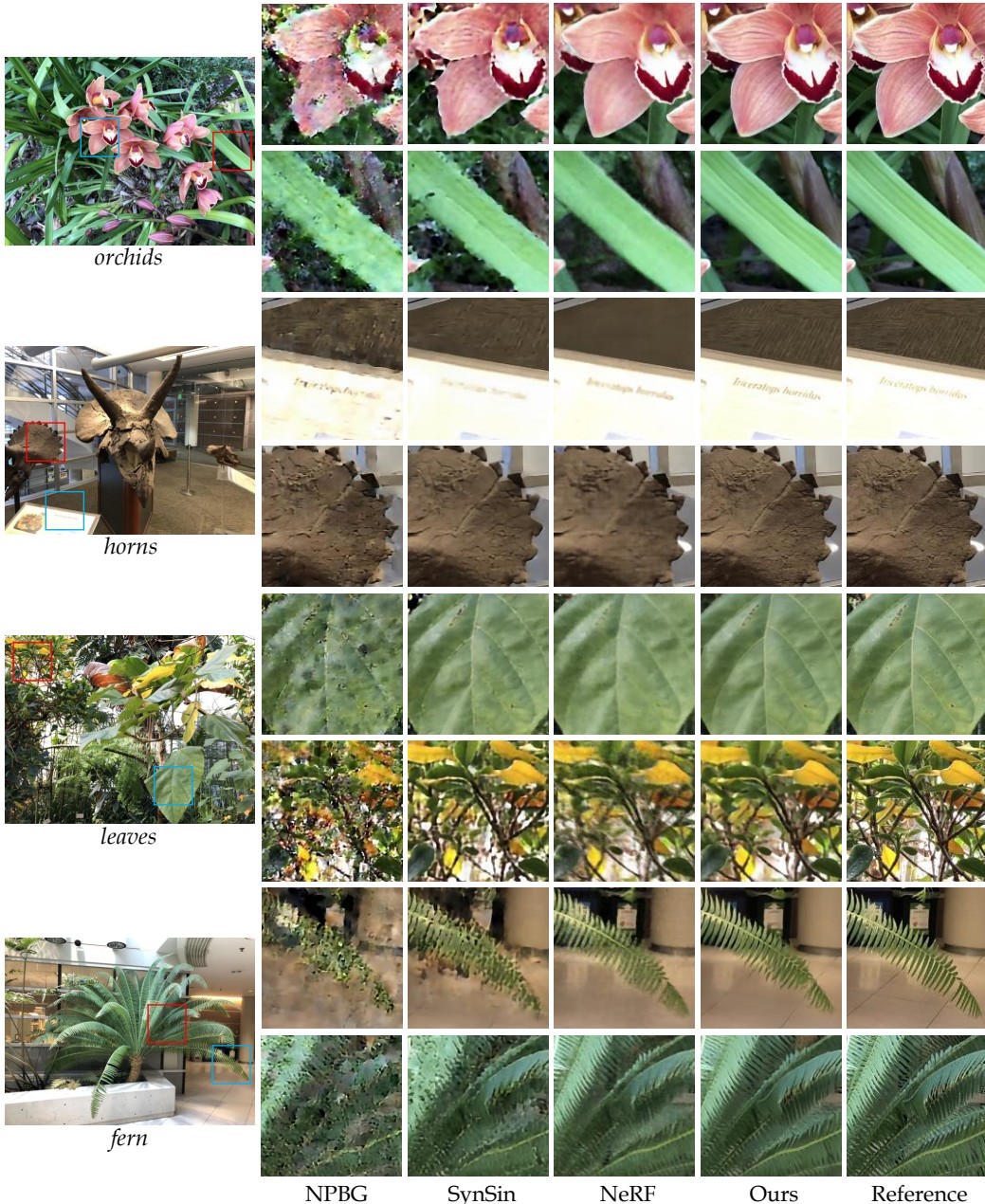

Figure 12: Compared to NeRF, our model can reconstruct very fine details such as the thin leaves in the *fern* scene; the cracks on the bones, the letters on the board, and the textures on the carpet in the *horns* scene; the strips on the flowers and leaves in the *orchids* and the *leaves* scene. Our method also has significantly better visual quality compared to the two point-based baselines.

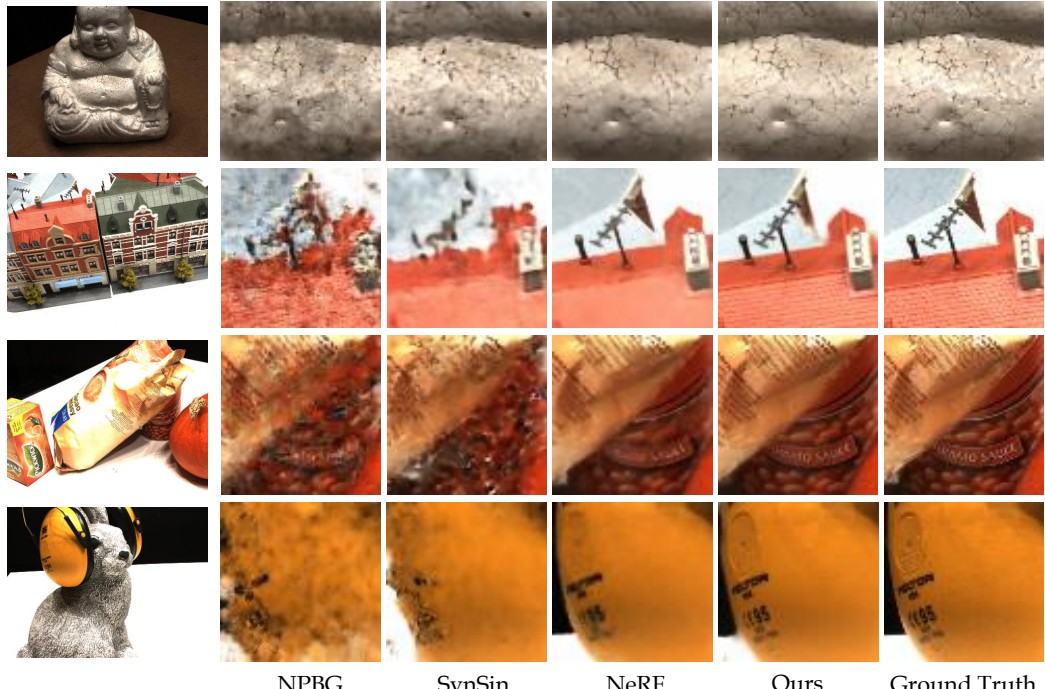

NPBG          SynSin          NeRF          Ours          Ground Truth

Figure 13: Our model can capture fine details, such as the cracks on the statue, the antenna and tiles on the roof, and the letters on the soup can and the earphone. NeRF results are often over-smoothed. The NPBG results contain some degrees of detail but are noisy, while fine details are often missing in the SynSin results.

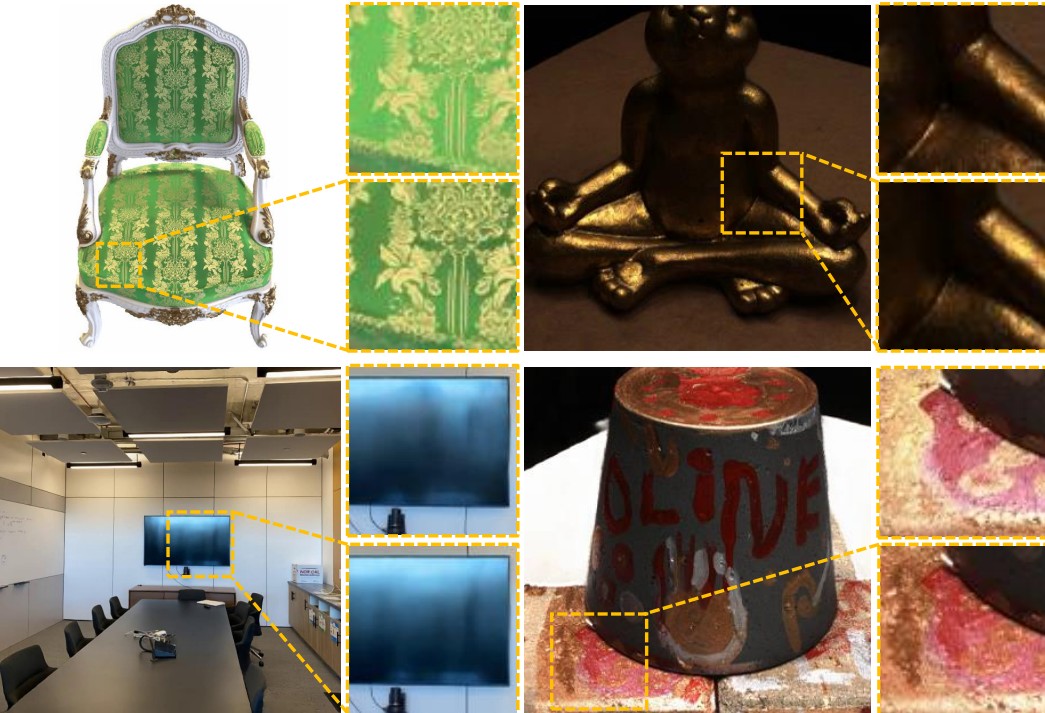

Figure 14: Qualitative results of the proposed spherical harmonics view-dependent shader. Two crops are rendered using the same camera pose and two different virtual viewing directions. Results show that our model can effectively learn the appearance of highly non-Lambertian surfaces.

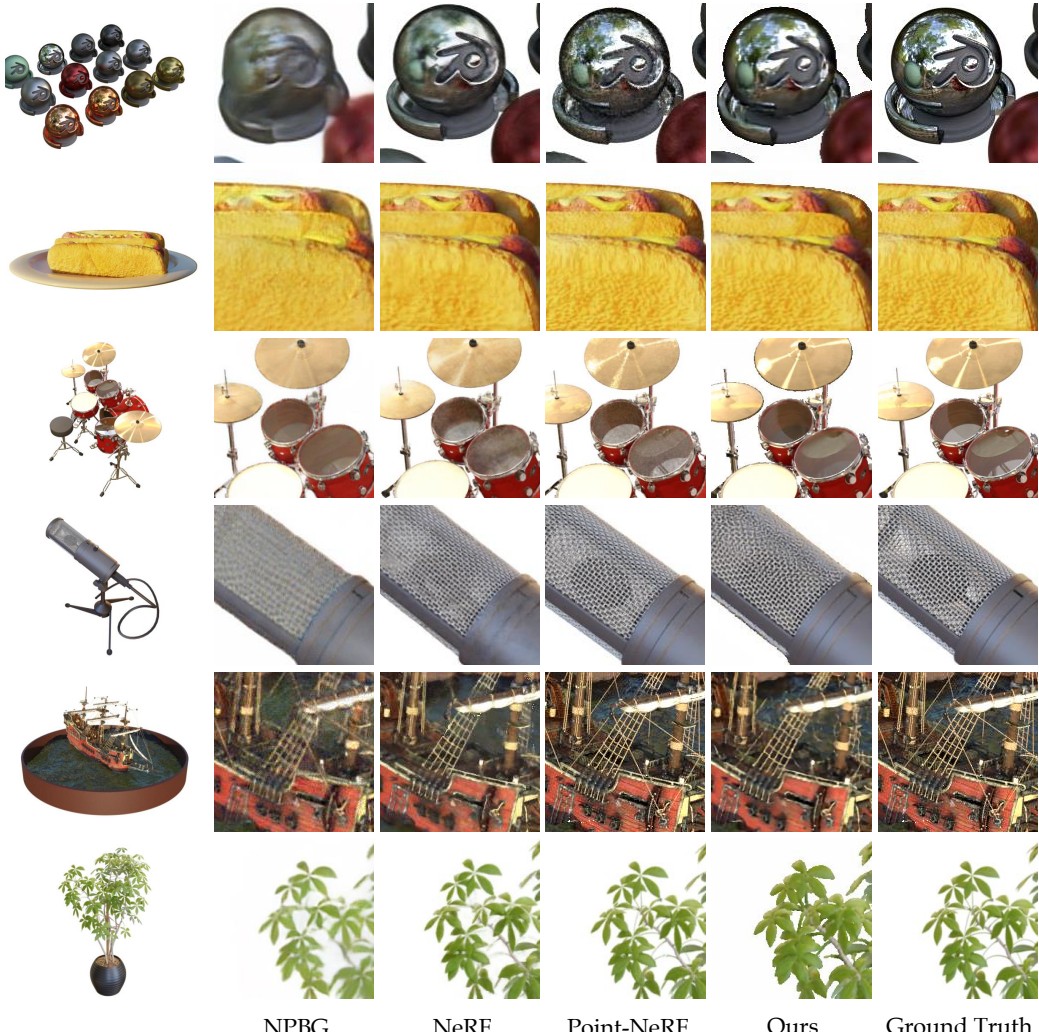

NPBG          NeRF          Point-NeRF          Ours          Ground Truth

Figure 15: Qualitative comparison of our method against baselines on the NeRF-Synthetic dataset. Our method is especially good at capturing surfaces with complex textures and strong non-Lambertian effects, such as the glossy ball and the transparent drum surfaces. For tiny structures such as the microphone and the ficus leaves, our rendering tends to be over-smoothed compared to Point-NeRF.

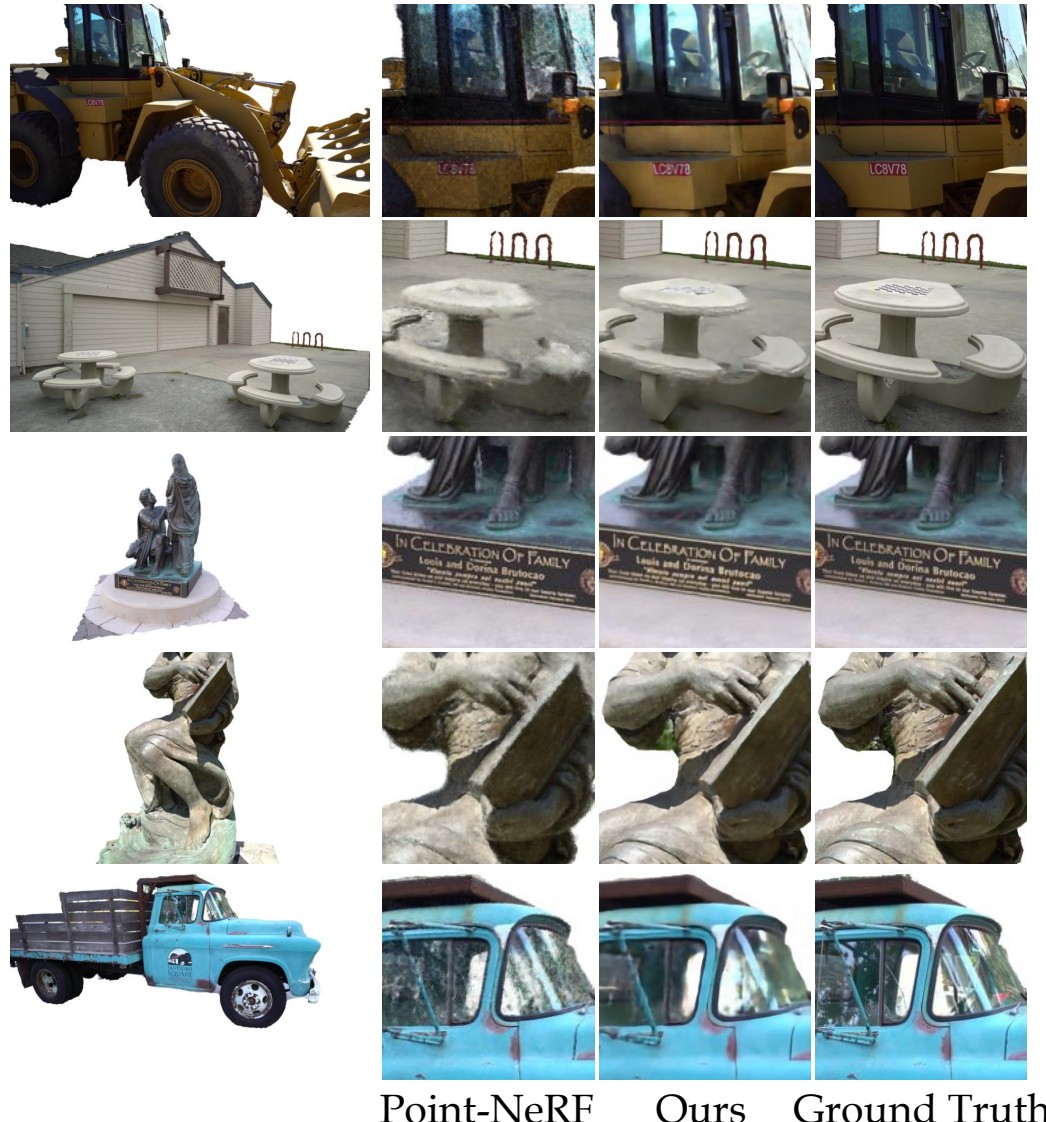

Point-NeRF    Ours    Ground Truth

Figure 16: Qualitative comparison of our model and Point-NeRF Xu et al. (2022) on the Tanks&Temples dataset. Our model renders smoother results in general, especially on reflective or transparent surfaces (*e.g.*, the windows of the truck).

# G PER-SCENE BREAKDOWN

We show the per-scene PSNR, SSIM and LPIPS in Tab. 10, 11, 12, 13, and 14.

| PSNR↑ | | | | | | | | | | |
|---|---|---|---|---|---|---|---|---|---|---|
| | 1 | 4 | 15 | 24 | 32 | 33 | 49 | 110 | 114 | 118 |
| NPBG | 16.91 | 18.07 | 17.54 | 18.58 | 17.62 | 15.47 | 17.48 | 22.87 | 22.49 | 26.75 |
| SynSin | 19.69 | 18.22 | 19.16 | 19.86 | 17.81 | 15.57 | 19.46 | 26.54 | 25.41 | 28.63 |
| NeRF | **28.43** | **26.74** | **25.79** | **28.03** | **26.87** | **26.49** | **28.35** | **31.34** | **29.10** | **38.55** |
| Ours | 24.90 | 25.15 | 25.08 | 24.98 | 25.21 | 23.26 | 25.00 | 30.47 | 29.03 | 33.74 |

| SSIM↑ | | | | | | | | | | |
|---|---|---|---|---|---|---|---|---|---|---|
| | 1 | 4 | 15 | 24 | 32 | 33 | 49 | 110 | 114 | 118 |
| NPBG | 0.576 | 0.558 | 0.640 | 0.615 | 0.644 | 0.604 | 0.687 | 0.733 | 0.711 | 0.756 |
| SynSin | 0.681 | 0.638 | 0.698 | 0.639 | 0.726 | 0.629 | 0.760 | 0.813 | 0.771 | 0.789 |
| NeRF | 0.818 | 0.737 | 0.830 | 0.793 | 0.878 | 0.888 | 0.888 | 0.878 | 0.841 | **0.906** |
| Ours | **0.851** | **0.845** | **0.901** | **0.862** | **0.894** | **0.898** | **0.893** | **0.900** | **0.889** | 0.904 |

| LPIPS↓ | | | | | | | | | | |
|---|---|---|---|---|---|---|---|---|---|---|
| | 1 | 4 | 15 | 24 | 32 | 33 | 49 | 110 | 114 | 118 |
| NPBG | 0.411 | 0.417 | 0.410 | 0.407 | 0.429 | 0.476 | 0.423 | 0.391 | 0.373 | 0.384 |
| SynSin | 0.322 | 0.345 | 0.333 | 0.334 | 0.329 | 0.348 | 0.340 | 0.324 | 0.341 | 0.351 |
| NeRF | 0.236 | 0.341 | 0.242 | 0.253 | 0.180 | 0.169 | 0.212 | 0.370 | 0.369 | 0.283 |
| Ours | **0.159** | **0.175** | **0.120** | **0.132** | **0.147** | **0.154** | **0.151** | **0.175** | **0.172** | **0.176** |

Table 10: Per-scene quantitative results on the DTU dataset.

| PSNR↑ | | | | | | | | |
|---|---|---|---|---|---|---|---|---|
| | Room | Fern | Fortress | Leaves | Orchids | Flower | T-Rex | Horns |
| NPBG | 22.75 | 19.33 | 24.35 | 15.71 | 15.95 | 21.06 | 19.86 | 20.79 |
| SynSin | 26.81 | 20.37 | 28.04 | 16.80 | 16.43 | 25.03 | 21.59 | 23.64 |
| NeRF | **32.70** | **25.17** | **31.16** | **20.92** | **20.36** | 27.40 | **26.80** | **27.45** |
| Ours | 30.35 | 23.80 | 30.88 | 18.76 | 20.14 | **27.75** | 24.73 | 26.14 |

| SSIM↑ | | | | | | | | |
|---|---|---|---|---|---|---|---|---|
| | Room | Fern | Fortress | Leaves | Orchids | Flower | T-Rex | Horns |
| NPBG | 0.813 | 0.569 | 0.766 | 0.441 | 0.397 | 0.654 | 0.686 | 0.668 |
| SynSin | 0.883 | 0.621 | 0.836 | 0.548 | 0.450 | 0.783 | 0.768 | 0.751 |
| NeRF | **0.948** | **0.792** | 0.881 | **0.690** | 0.641 | 0.827 | **0.880** | 0.828 |
| Ours | 0.941 | 0.784 | **0.901** | 0.649 | **0.677** | **0.859** | 0.870 | **0.853** |

| LPIPS↓ | | | | | | | | |
|---|---|---|---|---|---|---|---|---|
| | Room | Fern | Fortress | Leaves | Orchids | Flower | T-Rex | Horns |
| NPBG | 0.433 | 0.500 | 0.343 | 0.480 | 0.557 | 0.401 | 0.459 | 0.459 |
| SynSin | 0.328 | 0.405 | 0.237 | 0.384 | 0.473 | 0.258 | 0.355 | 0.366 |
| NeRF | **0.178** | 0.280 | 0.171 | 0.316 | 0.321 | 0.219 | **0.249** | 0.268 |
| Ours | 0.231 | **0.230** | **0.149** | **0.289** | **0.248** | **0.185** | 0.255 | **0.242** |

Table 11: Per-scene quantitative results on the LLFF dataset.

PSNR↑

| | Chair | Drums | Ficus | Hotdog | Lego | Mat. | Mic | Ship |
|---|---|---|---|---|---|---|---|---|
| NPBG | 26.47 | 21.53 | 24.60 | 29.01 | 24.84 | 21.58 | 26.62 | 21.83 |
| NPBG++ | - | - | 24.61 | 32.31 | - | - | 29.08 | - |
| NeRF | 33.00 | 25.01 | 30.13 | 36.18 | 32.54 | **29.62** | 32.91 | 28.65 |
| Point-NeRF | **35.40** | **26.06** | **36.13** | **37.30** | **35.04** | 29.61 | **35.95** | **30.97** |
| Ours | 30.49 | 22.78 | 25.43 | 33.24 | 27.94 | 26.02 | 28.80 | 25.07 |

SSIM↑

| | Chair | Drums | Ficus | Hotdog | Lego | Mat. | Mic | Ship |
|---|---|---|---|---|---|---|---|---|
| NPBG | 0.939 | 0.904 | 0.940 | 0.964 | 0.923 | 0.887 | 0.959 | 0.866 |
| NPBG++ | - | - | 0.925 | 0.964 | - | - | 0.967 | - |
| NeRF | 0.967 | 0.925 | 0.964 | 0.974 | 0.961 | 0.949 | 0.980 | 0.856 |
| Point-NeRF | **0.991** | **0.954** | **0.993** | **0.991** | **0.988** | **0.971** | **0.994** | **0.942** |
| Ours | 0.962 | 0.913 | 0.933 | 0.977 | 0.949 | 0.939 | 0.972 | 0.866 |

LPIPS↓

| | Chair | Drums | Ficus | Hotdog | Lego | Mat. | Mic | Ship |
|---|---|---|---|---|---|---|---|---|
| NPBG | 0.085 | 0.112 | 0.078 | 0.075 | 0.119 | 0.134 | 0.060 | 0.210 |
| NPBG++ | - | - | 0.070 | 0.050 | - | - | 0.029 | - |
| NeRF | 0.046 | 0.091 | 0.044 | 0.121 | 0.050 | 0.063 | 0.028 | 0.206 |
| Point-NeRF | **0.023** | **0.078** | **0.022** | 0.037 | **0.024** | **0.072** | **0.014** | **0.124** |
| Ours | 0.049 | 0.081 | 0.050 | **0.036** | 0.057 | **0.072** | 0.025 | 0.167 |

Table 12: Per-scene quantitative results on the NeRF-Synthetic dataset.

PSNR↑

| | Ignatius | Truck | Barn | Caterpillar | Family |
|---|---|---|---|---|---|
| NV | 26.54 | 21.71 | 20.82 | 20.71 | 28.72 |
| NeRF | 25.43 | 25.36 | 24.05 | 23.75 | 30.29 |
| NSVF | 27.91 | 26.92 | 27.16 | 26.44 | 33.58 |
| Point-NeRF | 28.43 | **28.22** | 29.15 | 27.00 | **35.27** |
| Ours | **29.62** | 28.05 | **29.80** | **27.37** | 34.07 |

SSIM↑

| | Ignatius | Truck | Barn | Caterpillar | Family |
|---|---|---|---|---|---|
| NV | 0.992 | 0.793 | 0.721 | 0.819 | 0.916 |
| NeRF | 0.920 | 0.860 | 0.750 | 0.860 | 0.932 |
| NSVF | 0.930 | 0.895 | 0.823 | 0.900 | 0.954 |
| Point-NeRF | 0.961 | **0.950** | **0.937** | **0.934** | **0.986** |
| Ours | **0.968** | 0.931 | 0.915 | 0.919 | 0.979 |

LPIPS$_{Alex}$↓

| | Ignatius | Truck | Barn | Caterpillar | Family |
|---|---|---|---|---|---|
| NV | 0.117 | 0.312 | 0.479 | 0.280 | 0.111 |
| NeRF | 0.111 | 0.192 | 0.395 | 0.196 | 0.098 |
| NSVF | 0.106 | 0.148 | 0.307 | 0.141 | 0.063 |
| Point-NeRF | 0.069 | **0.077** | 0.120 | **0.111** | 0.024 |
| Ours | **0.038** | 0.096 | **0.109** | 0.135 | **0.018** |

Table 13: Per-scene quantitative results on the Tanks&Temples dataset.

PSNR↑

| | 1 | 4 | 15 | 24 | 32 | 33 | 49 | 110 | 114 | 118 |
|---|---|---|---|---|---|---|---|---|---|---|
| Use DCC Filtering | 13.03 | 16.10 | 20.81 | 19.44 | 17.76 | 13.87 | 16.61 | 27.74 | 26.14 | 28.16 |
| No Adding; No Pruning | 23.47 | 23.51 | 23.84 | 23.37 | 23.31 | 21.93 | 23.51 | 28.97 | 27.76 | 30.88 |
| No Adding | 25.06 | 25.33 | 24.98 | 24.60 | 25.10 | 23.19 | 24.78 | 28.65 | 28.43 | 31.42 |
| No Gradient-based Refine | 24.99 | 25.13 | 24.72 | 24.90 | 25.00 | 22.99 | 24.75 | 30.24 | 28.88 | 33.61 |
| No View Dependence | 24.07 | 24.52 | 24.07 | 23.49 | 24.45 | 22.70 | 23.63 | 28.81 | 28.02 | 32.95 |
| View Dependence w/ MLP | 24.56 | 24.96 | 24.53 | 24.50 | 24.74 | 23.06 | 24.05 | 29.96 | 28.91 | 33.70 |
| No Point Dropout | 24.05 | 23.85 | 24.34 | 23.81 | 23.97 | 22.70 | 23.96 | 29.40 | 25.21 | 32.75 |
| Low Dropout Rate | 24.75 | 25.09 | 24.63 | 24.78 | 24.97 | 23.01 | 24.67 | 30.25 | 28.95 | 33.61 |
| High Dropout Rate | 24.79 | 25.07 | 24.94 | 24.43 | 25.11 | 23.02 | 24.62 | 30.17 | 28.88 | 33.57 |
| BatchNorm in UNet | 22.68 | 23.35 | 23.79 | 23.88 | 23.59 | 22.67 | 23.70 | 28.57 | 27.67 | 31.99 |
| InstanceNorm in UNet | 24.89 | 24.83 | 23.56 | 24.76 | 25.03 | 21.72 | 24.52 | 30.01 | 28.50 | 32.96 |
| 2-layer 1×1 Conv, no UNet | 15.57 | 18.46 | 17.84 | 17.24 | 16.91 | 15.13 | 15.38 | 26.46 | 25.35 | 27.99 |
| Complete Model | 24.90 | 25.15 | 25.08 | 24.98 | 25.21 | 23.26 | 25.00 | 30.47 | 29.03 | 33.74 |

SSIM↑

| | 1 | 4 | 15 | 24 | 32 | 33 | 49 | 110 | 114 | 118 |
|---|---|---|---|---|---|---|---|---|---|---|
| Use DCC Filtering | 0.772 | 0.790 | 0.888 | 0.846 | 0.861 | 0.824 | 0.850 | 0.872 | 0.862 | 0.870 |
| No Adding; No Pruning | 0.799 | 0.798 | 0.859 | 0.817 | 0.837 | 0.850 | 0.844 | 0.859 | 0.844 | 0.855 |
| No Adding | 0.853 | 0.847 | 0.902 | 0.868 | 0.897 | 0.898 | 0.894 | 0.886 | 0.887 | 0.892 |
| No Gradient-based Refine | 0.846 | 0.840 | 0.898 | 0.861 | 0.889 | 0.894 | 0.890 | 0.898 | 0.882 | 0.904 |
| No View Dependence | 0.837 | 0.832 | 0.890 | 0.857 | 0.889 | 0.892 | 0.887 | 0.890 | 0.884 | 0.901 |
| View Dependence w/ MLP | 0.845 | 0.838 | 0.897 | 0.862 | 0.893 | 0.895 | 0.889 | 0.898 | 0.890 | 0.906 |
| No Point Dropout | 0.815 | 0.870 | 0.817 | 0.834 | 0.854 | 0.865 | 0.859 | 0.870 | 0.856 | 0.876 |
| Low Dropout Rate | 0.848 | 0.841 | 0.895 | 0.858 | 0.890 | 0.893 | 0.890 | 0.897 | 0.885 | 0.901 |
| High Dropout Rate | 0.851 | 0.843 | 0.899 | 0.860 | 0.895 | 0.895 | 0.892 | 0.900 | 0.889 | 0.904 |
| BatchNorm in UNet | 0.822 | 0.791 | 0.883 | 0.839 | 0.850 | 0.878 | 0.869 | 0.876 | 0.870 | 0.890 |
| InstanceNorm in UNet | 0.849 | 0.841 | 0.873 | 0.856 | 0.888 | 0.848 | 0.879 | 0.885 | 0.881 | 0.894 |
| 2-layer 1×1 Conv, no UNet | 0.627 | 0.624 | 0.694 | 0.604 | 0.657 | 0.510 | 0.560 | 0.765 | 0.749 | 0.767 |
| Complete Model | 0.851 | 0.845 | 0.901 | 0.862 | 0.894 | 0.898 | 0.893 | 0.900 | 0.889 | 0.904 |

LPIPS↓

| | 1 | 4 | 15 | 24 | 32 | 33 | 49 | 110 | 114 | 118 |
|---|---|---|---|---|---|---|---|---|---|---|
| Use DCC Filtering | 0.217 | 0.238 | 0.135 | 0.146 | 0.175 | 0.223 | 0.193 | 0.205 | 0.211 | 0.215 |
| No Adding; No Pruning | 0.205 | 0.220 | 0.169 | 0.185 | 0.208 | 0.213 | 0.200 | 0.198 | 0.200 | 0.209 |
| No Adding | 0.167 | 0.177 | 0.125 | 0.135 | 0.146 | 0.150 | 0.154 | 0.195 | 0.183 | 0.197 |
| No Gradient-based Refine | 0.158 | 0.176 | 0.123 | 0.133 | 0.149 | 0.153 | 0.154 | 0.175 | 0.174 | 0.178 |
| No View Dependence | 0.165 | 0.179 | 0.127 | 0.132 | 0.152 | 0.155 | 0.154 | 0.179 | 0.178 | 0.179 |
| View Dependence w/ MLP | 0.169 | 0.179 | 0.129 | 0.136 | 0.152 | 0.152 | 0.156 | 0.177 | 0.177 | 0.177 |
| No Point Dropout | 0.199 | 0.211 | 0.160 | 0.166 | 0.193 | 0.207 | 0.187 | 0.194 | 0.197 | 0.194 |
| Low Dropout Rate | 0.161 | 0.177 | 0.124 | 0.132 | 0.150 | 0.151 | 0.151 | 0.175 | 0.176 | 0.178 |
| High Dropout Rate | 0.158 | 0.172 | 0.121 | 0.131 | 0.147 | 0.155 | 0.153 | 0.177 | 0.176 | 0.180 |
| BatchNorm in UNet | 0.180 | 0.192 | 0.136 | 0.146 | 0.164 | 0.160 | 0.159 | 0.190 | 0.188 | 0.191 |
| InstanceNorm in UNet | 0.161 | 0.179 | 0.150 | 0.139 | 0.160 | 0.196 | 0.161 | 0.182 | 0.179 | 0.186 |
| 2-layer 1×1 Conv, no UNet | 0.333 | 0.346 | 0.328 | 0.359 | 0.361 | 0.446 | 0.418 | 0.315 | 0.317 | 0.332 |
| Complete Model | 0.159 | 0.175 | 0.120 | 0.132 | 0.147 | 0.154 | 0.151 | 0.175 | 0.172 | 0.176 |

Table 14: Per-scene breakdown for the ablation studies on the DTU dataset.

