# OpenReview forum: "View Synthesis with Sculpted Neural Points"
_ICLR.cc/2023/Conference — ICLR 2023 notable top 5%_

### Official Review · Reviewer_bjQn · 2022-10-22

**Confidence:** 4
**Correctness:** 4
**Technical Novelty And Significance:** 3
**Empirical Novelty And Significance:** 3
**Recommendation:** 8

**Clarity, Quality, Novelty And Reproducibility:**

Overall this paper is clearly written and easy to follow. The proposed point sculpting, though sounds a bit engineering, is effective in improving the completness of the reconstruction. The proposed point dropout layer is also sound and novel, and is effective in improving the generalization to novel views. SOTA results have been reported. Sufficient details have been provided for reproduction of the results.

**Strength And Weaknesses:**

Strength
+ The proposed point sculpting can effectively improve the completness of the reconstruction.
+ The proposed point dropout layer can alleviate the over-fitting problem by giving all points a chane to get optimized even if they are covered by others in the input views.
+ The proposed high-dimensional spherical harmonics point features can effectively capture non-Lamberian visual effects.
+ The proposed method is 100 times faster than the popular NeRF approach in rendering.
+ The proposed method can produce better visual quality than NeRF and other point-based methods.
+ The proposed method achieves SOTA in terms of SSIM and LPIPS metrics.
+ The proposed method allows easy fine-grained scene editing due to its point-based representation.

Weaknesses
- Examples in the Appendix show that the proposed method is good at capturing complex textures and strong non-Lambertian effects, but tends to produce over-smoothed results for tiny structures. This may suggest that the proposed spherical harmonics point features are effective in modeling the view-dependent appearance of the scene but the point-based representation may not be good (dense) enough to  capture fine scene geometry. The over-smoothed results may also be related to the use of U-Net for final rendering. Finally, the averaging of 2 results generated from 2 random point subsets may also contribute to the problem. More analysis should be carried out regarding this issue.
- In 3.2, it was mentioned that U-Net was employed to convert the 2D feature map F of a target veiw into RGB image. However, in 4.2 (point adding step), it was mentioned that the point features were treated directly as RGB values during rasterization without using U-Net in optimizing the existing point.  Please clarify how can the features be treated directly as RGB values in this step.

**Summary Of The Paper:**

This paper addresses the problem of novel view synthesis from a set of input images. The authors presented a point-based method which is build on top of exsiting works on differentiable point-based rendering. They introduced point sculpting for pruning and adding points to improve the photo-consistency and completness of the reconstruction. They also exploited sphereical harmnonics in high-dimensionalpoint feature space to capture non-Lambertain visual effects. They proposed a point dropout layer that can significantly improves the generalization to novel views. Experimental results show that their method can achieve better visual quality than the popular NeRF approach and at the same time being 100 times faster in rendering speed.

**Summary Of The Review:**

The improvements introduced to the point-based approach for novel view synthesis are novel and effective. Comprehensive evaluations are presented, showing SOTA results over existing NeRF and point-based methods. I would support accepting this paper.

---

> ### Author Response · Authors · 2022-11-16
> **Response to reviewer bjQn**
>
> We would like to thank the reviewer for their time and detailed comments. Here are the replies to your questions and concerns:
>
> &nbsp;
>
> **Q1:**
> > The paper lacks analysis on the over-smoothed results. Is this issue related to 1) point-based representation may not be good (dense) enough to capture fine scene geometry? 2) using a U-Net? 3) averaging of 2 results generated from 2 random point subsets?
>
>
> **A1:**
> We agree the point cloud quality may not be accurate enough when the geometry is too fine, and we have pointed out this limitation in the NeRF-Synthetic experiments in Sec.5.2 in the original paper.
>
> We agree that the U-Net adds smoothness to the output image, but we find it crucial to getting good rendering results. A tradeoff exists here but additional experiments show such smoothness regularization is more helpful than harmful in the current system (PSNR 19.63 v.s. 26.68 for 1x1 Conv v.s. U-Net, See Tab.4. in the revision). We have now added more discussions in Sec.3.2 for the intuitions behind using a U-Net.
>
> Experiments show that averaging more subsets does NOT contribute to the over-smoothing problem, because adding more subsets does not hurt the performance, as shown in an additional experiment in Appendix.E in the revision.
>
> &nbsp;
>
> **Q2:**
> > How can the features be treated directly as RGB values in the point adding step, without using a U-Net?
>
> **A2:**
> We use $f_i \in \mathbb{R}^{27}$, which is converted to $s_i \in \mathbb{R}^{3}$ by the SH layer. $s_i$ is directly treated as RGB values during rasterization. We have now clarified this in Sec.4.2 in the revision.

---

### Official Review · Reviewer_kuEi · 2022-10-25

**Confidence:** 5
**Correctness:** 3
**Technical Novelty And Significance:** 3
**Empirical Novelty And Significance:** 3
**Recommendation:** 6

**Clarity, Quality, Novelty And Reproducibility:**

This paper is good and clear in writing. The ideas proposed are novel. The authors have provided abundant information to reproduce the work.

**Strength And Weaknesses:**

Strength:

- The proposed point sculpting technique is novel. The method achieves good performance on all benchmarked datasets and has faster training and inference speed when compared to NeRF or other point-based baselines.

- Using spherical harmonics in the feature space and eliminating the need for a dir encoding MLP seems to be a novel but useful technique.

Weaknesses:

- According to Appendix B.2, the authors applied different MVS networks (at least different weights) when evaluating on different datasets. Especially, when evaluating on LLFF data, they finetune the MVS model scene-by-scene. This may cause concern/confusion because 1) the time for COLMAP and scene-by-scene fine-tuning should be taken into account when comparing, rendering the method less efficient for these scenes; 2) it is unclear why not using COLMAP point clouds directly. The authors should clarify these and ideally use a same generalized MVS model for a fair comparison.

- Pulsar is a very related direct baseline of this method. However, a comparison with Pulsar seems to be missing in this paper.

- The intuition behind employing a U-Net for rendering is unclear. Is it possible to use per-pixel MLP to render the features (MLP: feat -> RGB)? Is the U-Net pre-trained and weight-shared across datasets?

**Summary Of The Paper:**

The paper proposes a point-based rendering framework for novel view synthesis. The method starts with an initial point cloud estimation from MVS systems. With the proposed key technique named point sculpting, the framework is less prone to initial point cloud error and could add and remove points within the optimization process. The method uses point rasterization instead of volume rendering when synthesizing novel views. The paper includes extensive experiments showing good results.

**Summary Of The Review:**

Overall the method proposed is novel and effective. I am leaning toward acceptance due to their technique contributions and good performance.

---

> ### Author Response · Authors · 2022-11-16
> **Response to reviewer kuEi**
>
> We would like to thank the reviewer for their time and detailed comments. Here are the replies to your questions and concerns:
>
> &nbsp;
>
> **Q1.1:**
> > MVS model: the time for COLMAP and scene-by-scene fine-tuning should be taken into account when comparing.
>
> **A1.1:**
> We agree such overhead exists but it is not significant. On the LLFF dataset, running the COLAMP MVS takes about 0.3 hours and the finetuning takes about 1 hour. This additional overhead is insignificant compared to the 6-8 hours training time of our model. Taking this overhead into account, our method is still more efficient in training than NeRF (about 21 hours on LLFF). We have now clarified these in Appendix.B.2 in the revision.
>
> &nbsp;
>
> **Q1.2:**
> > MVS model: why not use COLMAP point clouds directly?
>
> **A1.2:**
> The learning-based MVS model we use works better than the COLMAP pointcloud. When we train our differentiable rendering pipeline directly with the COLMAP fused point cloud on the LLFF dataset, the resulting PSNR, SSIM, and LPIPS are 24.81/0.817/0.240, which are worse than the 25.32/0.817/0.229 by our method. This is due to the inferior completeness of the COLMAP point cloud, especially in textureless areas such as the ceiling. We have now added more results to Appendix.B.2 in the revision.
>
> &nbsp;
>
> **Q1.3:**
> > MVS model: why not use the same generalized MVS network across all datasets?
>
> **A1.3:**
> Using different MVS network weights for different datasets can achieve better results than using a single MVS network. In practice, it is reasonable to take advantage of training data available from a particular application domain to maximize domain-specific performance. That said, we agree that using a single network across all datasets is a more elegant solution, but developing such a generalizable MVS system is the goal of MVS research and is beyond the scope of this work.  MVS research has been advancing rapidly, and we are optimistic that in the near future a generalized MVS would perform sufficiently well to make domain-specific training unnecessary.
>
> &nbsp;
>
> **Q2:**
> > A comparison with Pulsar is missing in this paper.
>
> **A2:**
> We agree Pulsar is a highly related work. Unfortunately, we are unable to do a quantitative comparison with Pulsar at this moment, because we have contacted the authors and they were not planning to release the train/eval code for the view synthesis section of the paper. We do show qualitatively that our method is better than Pulsar, see Appendix.D in the revision. Conceptually, while using the same rasterization backbone, our method is better because 1) we use MVS to initialize point positions, whereas they start from random positions; 2) we do point adding to improve the completeness, whereas they don’t; 3) we use SH features and a point dropout layer to boost the performance, whereas they don’t have such designs.
>
> &nbsp;
>
> **Q3:**
> > What is the intuition behind employing a U-Net for rendering? Is it possible to use per-pixel MLP to render the features? Is the U-Net pre-trained and weight-shared across datasets?
>
> **A3:**
> The usage of U-Net follows previous work NPBG[1]. The intuition behind using a U-Net is that it can remove tiny holes and noise in the feature map. As we use the dropout layer and relatively small point radius, the rasterized feature map often contains tiny holes. The U-Net architecture provides large receptive fields, which has been shown desirable in image inpainting literature[2]. We have now added more discussions regarding U-Net in Sec.3.2 in the revision.
>
> Replacing U-Net with MLP leads to worse results. We add an ablation experiment where we use a 2-layer MLP (implemented with 1x1 Conv layers). Experiments results on the DTU dataset show that such a design is significantly worse than UNet (PSNR 19.63 v.s. 26.68). See Tab.4 in the revision for details.
>
> The UNet is trained from scratch for each scene individually. We have clarified this in Sec.5.1 in the original paper: “We initialize the U-Net parameters randomly”. We have now emphasized that again in Appendix.B.3 in the revision.
>
> &nbsp;
>
> **References:**
>
> [1] Aliev, Kara-Ali, et al. "Neural point-based graphics." Proceedings of the European conference on computer vision (ECCV). 2020.
>
> [2] Liu, Guilin, et al. "Image inpainting for irregular holes using partial convolutions." Proceedings of the European conference on computer vision (ECCV). 2018.

---

### Official Review · Reviewer_UGSW · 2022-10-25

**Confidence:** 3
**Correctness:** 4
**Technical Novelty And Significance:** 2
**Empirical Novelty And Significance:** 3
**Recommendation:** 8

**Clarity, Quality, Novelty And Reproducibility:**

Clarity: The paper is clearly written and easy to follow.

Quality: The proposed method achieves view synthesis quality comparable to state-of-the-art while being much faster, which I suggest is good enough to be accepted.

Novelty: While the proposed method is largely built upon existing works, as pointed out by authors themselves, many new implementation improvements may be very useful for future researcher who hopes to use point-based representation.

Reproducibility: The code is attached to the submission.

**Details Of Ethics Concerns:**

I don't have any ethical concerns.

**Strength And Weaknesses:**

Strengths of the paper:
1. The paper is well written and easy to follow.

2. The contribution of the paper is clear and well-presented by authors. While this paper is built on existing works, author did a good job to combine them together and also proposed non-trivial implementation improvements that lead to higher quality results, which may be very useful for future researchers.

3. Experiments are convincing and comprehensive. Authors test their method on many widely used datasets.

Weaknesses of the paper:
1. The technical novelty may be somehow limited. But I believe this is not a very big issue.

2. Spherical Harmonic Point Feature: I have two questions here. First, if we only have degree 2, how could we handle very specular materials in DTU dataset? Second, why s_i's dimension is K/9? In which step do we turn feature vector to actual SH coefficient and compute the radiance?

3. Point Dropout Layer: Will random subsets cause flickering in different views? From the video, this does not happen. Do authors have any comments on why it won't happen?

4. Point sculpting: Can authors comment on how the proposed point pruning and adding method are different from those in Point-NeRF? This may help readers better understand the relationships between the 2 methods.

**Summary Of The Paper:**

This paper proposed a novel neural point-based rendering framework that can achieve view synthesis quality comparable to volume rendering based methods while being 100x faster. The method is based on existing approach but with non-trivial novel techniques that significantly improve the rendering quality, including SH features, point dropout layer, 2-D rendering without batch norm, point pruning and growing. Experiments on various

**Summary Of The Review:**

I believe this is a solid paper with clear contribution and improvements. While the technical novelties may be limited, a significant faster point-based neural rendering framework with high-quality can be empirically very useful. Authors also did comprehensive and convincing experiments to demonstrate the improvements of the proposed method. Therefore, I lean towards accepting this paper. I will really appreciate if authors can address my question in the weakness section. I will also learn from other reviewers's comments.

---

> ### Author Response · Authors · 2022-11-16
> **Response to reviewer UGSW**
>
> We would like to thank the reviewer for their time and detailed comments. Here are the replies to your questions and concerns:
>
> &nbsp;
>
> **Q2.1:**
> > Spherical Harmonic Point Feature: if we only have degree 2, how could we handle very specular materials in DTU dataset?
>
> **A2.1:**
> The choice of degree follows Plenoxels[1]. We don’t use higher degree SHs for two reasons: 1) 9 basis are empirically sufficient to capture highly specular surfaces. See supplementary video 4’51’’ for an example of the rabbit scene in the DTU dataset, where our method captures the metallic surfaces well. 2) Each basis corresponds to a free variable in the feature vector, therefore higher degree means more model parameters and higher memory cost, which is undesirable. We have now added additional discussions to Sec.3.2 in the revision.
>
> &nbsp;
>
> **Q2.2:**
> > Spherical Harmonic Point Feature: why s_i's dimension is K/9? In which step do we turn feature vector to actual SH coefficient and compute the radiance?
>
> **A2.2:**
> We first compute the SH basis according to $v$, yielding a basis vector $b_v \in \mathbb{R}^{9\times 1}$. We then reshape $f_i\in \mathbb{R}^{K}$ into $f_i' \in \mathbb{R}^{\frac{K}{9} \times 9}$, and finally compute $s_i$ with a dot product $s_i = f_i' \cdot b_v$. We have now clarified this in Sec.3.2 in the revision.
>
> &nbsp;
>
> **Q3:**
> > Point Dropout Layer: Will random subsets cause flickering in different views? Why won't this happen in the video results?
>
> **A3:**
> This is an excellent point. We observe such flickering if we use independent subsets for each frame. Instead, we do random subsample only once and use the same subsets across all frames, which leads to significantly better results. We have now clarified this technical detail in Sec.3.2 in the revision.
>
> &nbsp;
>
> **Q4:**
> > How is the proposed point pruning and adding method different from those in Point-NeRF?
>
> **A4:**
> While Point-NeRF and our method both refine the point cloud, the differences are that: 1) their pruning is based on the volume density trained at test time to optimize photoconsistency, while our point pruning is based on multiview consistency and doesn't require test-time training; 2) their point growing progressively adds points near existing points, while our point adding has only one round and can add new points in any location. We have now added the additional discussion to the related works section of the paper.
>
> &nbsp;
>
> **References**
>
> [1] Fridovich-Keil, Sara, et al. "Plenoxels: Radiance Fields Without Neural Networks." Proceedings of the IEEE/CVF Conference on Computer Vision and Pattern Recognition (CVPR). 2022.

---

### Decision · Program_Chairs · 2023-01-20

**Decision:**

Accept: notable-top-5%

**Justification For Why Not Higher Score:**

NA

**Justification For Why Not Lower Score:**

This paper receives 2x accept, good paper and 1x marginally above the acceptance threshold. All reviewers think that the proposed method is effective and the results are good.

**Metareview: Summary, Strengths And Weaknesses:**

This paper receives 2x accept, good paper and 1x marginally above the acceptance threshold. The reviewers agree that the proposed method leads to higher quality results, the paper is well written and the experiments are convincing and comprehensive. The weaknesses are mostly clarifications that are answered in the authors' responses.


**Note From Pc:**

if the above contains the word "oral" or "spotlight" please see: "oral" presentation means -> notable-top-5% and "spotlight" means -> notable-top-25%. As stated in our emails, we are disassociating presentation type from AC recommendations